# The regional scale surface mass balance of Pine Island Glacier, West Antarctica over the period 2005–2014, derived from airborne radar soundings and neutron probe measurements

Stefan Kowalewski[1], Veit Helm[1], Elizabeth Morris[2], and Olaf Eisen[1]

[1]Alfred Wegener Institute, Helmholtz Centre for Polar and Marine Research, Bremerhaven, Germany
[2]Scott Polar Research Institute, University of Cambridge, UK

**Correspondence:** Stefan Kowalewski (stefan.kowalewski@awi.de)

**Abstract.** We derive recent surface mass balance (SMB) estimates from airborne radar observations along the iSTAR traverse (2013,2014) at Pine Island Glacier (PIG), West Antarctica. Ground based neutron probe measurements provide information of snow and firn density with depth at 22 locations and were used to date internal annual reflection layers. The 2005 layer was traced for a total distance of 2367 km to determine annual mean SMB for the period 2005–2014. Using complementary SMB estimates from two regional climate models, RACMO2.3p2 and MAR, and a geostatistical kriging scheme, we determine a regional scale SMB distribution with similar main characteristics to that determined for the period 1985–2009 in previous studies. Local departures exist for the northern PIG slopes, where the orographic precipitation shadow effect appears to be more pronounced in our observations, and the southward interior, where the SMB gradient is more pronounced in previous studies. We derive total mass inputs of $79.9 \pm 19.2 \, \mathrm{Gt \, yr^{-1}}$ and $82.1 \pm 19.2 \, \mathrm{Gt \, yr^{-1}}$ to the PIG basin based on complementary ASIRAS–RACMO and ASIRAS–MAR SMB estimates, respectively. These are not significantly different to the value of $78.3 \pm 6.8 \, \mathrm{Gt \, yr^{-1}}$ for the period 1985–2009. Thus, there is no evidence of a secular trend at decadal scales in total mass input to the PIG basin. We note, however, that our estimated uncertainty is more than twice the uncertainty for the 1985–2009 estimate on total mass input. Our error analysis indicates that uncertainty estimates on total mass input are highly sensitive to the selected krige methodology and assumptions made on the interpolation error, which we identify as the main cause for the increased uncertainty range compared to the 1985–2009 estimates.

## 1 Introduction

The stability of the West Antarctic Ice Sheet (WAIS) is a major concern for scientists seeking to predict global sea level rise. Transport of heat from upwelling circumpolar deep water has proved to be a critical driver of Antarctic ice shelf thinning and grounding line retreat, thus initiating the acceleration of marine-terminating outlet glaciers (e.g. Hillenbrand et al., 2017). In particular the Amundsen Sea sector has experienced an unprecedented acceleration in ice discharge since the beginning of satellite based ice flow observations in the 1970s. Three quarters of this ice discharge stem from the Thwaites and Pine Island glaciers with both showing evidence of rapid acceleration since the 1970s (Mouginot et al., 2014) and spreading of surface lowering along their tributaries over the past two decades (Konrad et al., 2017). While spaceborne observations indicate that

this acceleration has levelled off recently (Rignot et al., 2019), they also support model projections suggesting modest changes in mass balance, i.e. the resulting net ice loss after accounting for all loss and gain processes, for the next decades to come (Bamber and Dawson, 2020). The dynamic ice loss is mainly responsible for the negative mass balance of Pine Island Glacier (PIG). The net input is commonly referred to as the surface mass balance (SMB), i.e. snowfall minus sublimation, meltwater runoff, and erosion/deposition of snow (Lenaerts et al., 2012; Medley et al., 2013). Various methods exist to measure the SMB on the ground (Eisen et al., 2008). The remoteness of WAIS makes such measurements logistically challenging, in particular when extending these measurements to regional scales. Basin wide total mass input estimates strongly depend on the coverage and quality of SMB measurements. The study of Medley et al. (2014), hereinafter abbreviated as ME14, presents the first comprehensive survey of mean annual SMB between 1985 and 2009/10 from airborne radar based observations of the Thwaites and Pine Island glaciers. The authors demonstrated that such airborne radar observations provide a critical means to overcome logistical challenges. However, these measurements rely on assumptions about the dielectric properties of snow and firn, which include knowledge of their vertical density profiles. In this sense, ground-truthing measurements remain an important tool for calibrating the radar soundings.

As part of the iSTAR Ice Sheet stability programme, a traverse across the Pine Island Glacier (PIG) was carried out in 2013/14 (T1) and repeated the year after (T2). In total 22 sites were occupied during both traverses. Boreholes of at least 13 m depth were drilled at each site during traverse T1. Density–depth profiles were measured with a Neutron Probe (NP) device during both traverses (Morris et al., 2017) and supplementary analysis of firn cores was performed for 10 sites during traverse T2 to determine additional independent proxies related to the annual snow accumulation (Konrad et al., 2019).

The Alfred Wegener Institute (AWI) contributed to the iSTAR traverse T2 with radar soundings from the Airborne SAR / Interferometric Radar Altimeter System (ASIRAS) aboard the Polar 5 research plane. Previous ASIRAS missions have demonstrated its capability to track annual snow accumulation layers of the upper firn column at regional scales over Greenland (Hawley et al., 2006; Overly et al., 2016). The PIG flight track connects all iSTAR sites so that internal annual snow accumulation layers can be traced to make regional scale SMB estimates. By comparison with earlier SMB measurements at PIG, the vertical profiling based on the ASIRAS soundings achieves a resolution that is one order of magnitude higher (Tab. 1), which helps to trace narrow internal snow accumulation layers. In addition, the ASIRAS flight track contains several crossovers which we used to validate the same isochronal reflector from different directions.

In this study we first address local departures between SMB estimates from ASIRAS and NP measurements to evaluate the uncertainty of our regional scale ASIRAS SMB estimates. We then compare our results with those reported by ME14 and discuss differences between both data sets. Finally, we apply our new regional scale SMB estimates to different PIG mass balance inventories to evaluate their impact in light of the current stability of the study area. We include a list of abbreviations and notations in Appendix A.

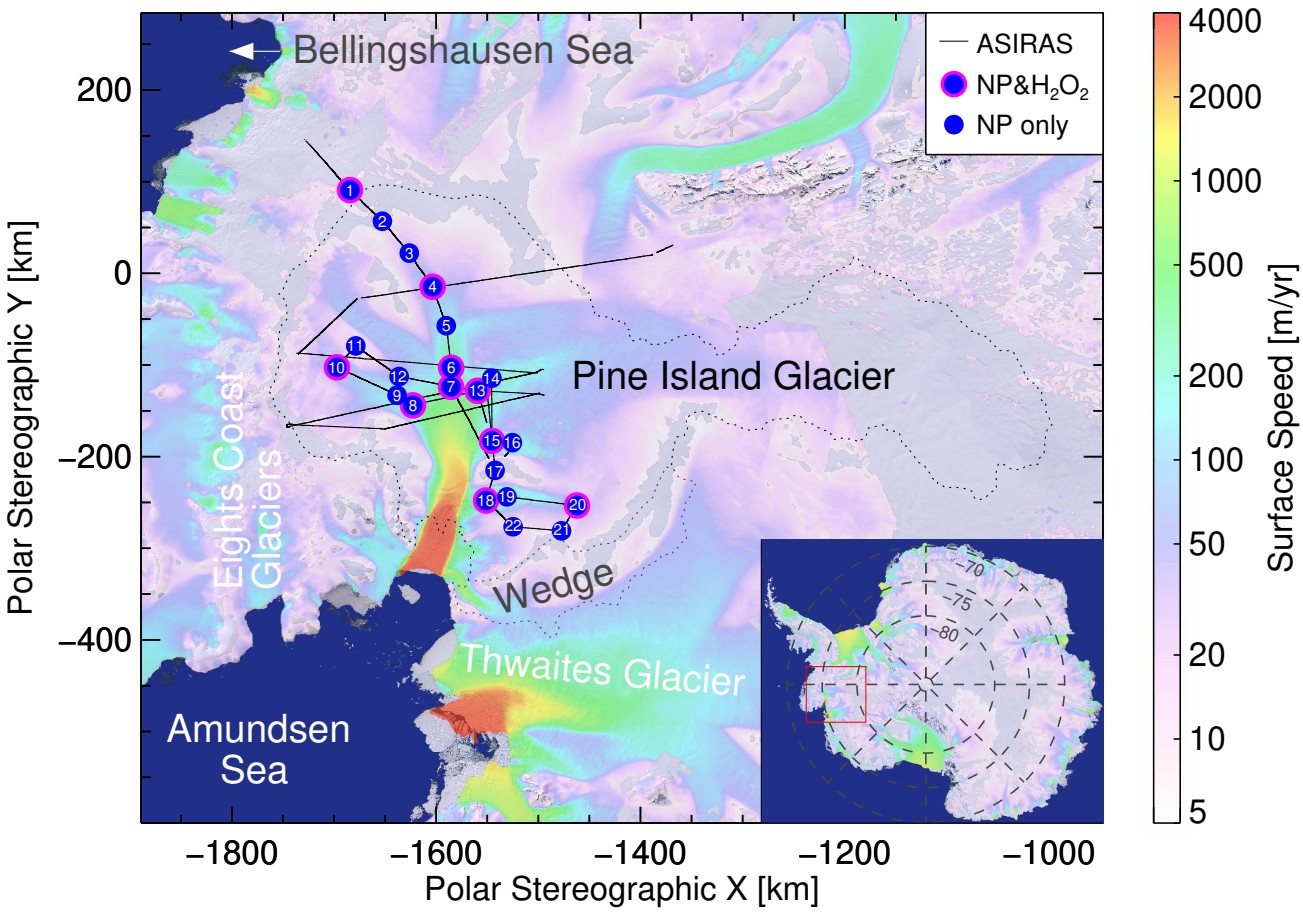

**Figure 1.** ASIRAS–iSTAR survey projected on polar stereographic coordinates: black lines denote the ASIRAS flight track, numbered blue circles the iSTAR sites with shallow ($\sim 13$ m) neutron probe snow density measurements, and magenta–blue circles iSTAR sites with additional deep ($\sim 50$ m) firn core measurements of $H_2O_2$ during traverse T2. Surface flow speeds from Rignot et al. (2017) are overlayed by colour shadings on top of Landsat imagery (U.S., Geological Survey, 2007). Dotted lines denote basin outlines based on Fretwell et al. (2013) analysis (data accessed via SCAR Antarctic Digital Database on 23 April 2019).

## 2 Data and methods

### 2.1 iSTAR traverse

The iSTAR traverse followed the PIG main trunk as well as its tributaries as shown in Fig. 1. A total flight track (black lines) of 2486 km was covered by the ASIRAS measurements between 1 and 3 December 2014. Following ME14, the basin outlines (dashed lines) include the Wedge zone between PIG and Thwaites. The main emphasis of the iSTAR campaign was on the fast flowing segments of PIG, thus we lack measurements from the southward interior. Earlier observations from ME14 suggest

| radar system | ASIRAS | CReSIS Accu-R | pulseEKKO PRO |
| --- | --- | --- | --- |
| | (this study) | (M14) | (Konrad et al., 2019) |
| operation | airborne | airborne | ground based |
| SMB averaging period | 2005–2014 | 1985–2009 | 1986–2014 |
| density profiles | iSTAR NP | ITASE and 2010 cores | iSTAR cores |
| dating markers | density guided | $H_2O_2$, water isotope ratios, | $H_2O_2$ |
| | with $H_2O_2$ | non-sea-salt-sulfur to sodium ratio | |
| vertical range bin (firn) | $7.3 \pm 0.3$ cm | 62 cm | 100 cm |
| along track bin | 4.5 m | order of 10 m | 1.4 m |
| maximum sampling depth | 30 m | 300 m | (90-120) m |

**Table 1.** Approximate sample bin resolution and maximum depth of: SAR level_1b processed ASIRAS data with indicated standard deviation of the vertical range bin based on the two-way-travel time (TWT) to depth conversion of this study, CreSIS Accumulation radar according to M14, and pulseEKKO PRO GPR discussed in Konrad et al. (2019). For the GPR system we estimate the maximum sampling depths based on shared radargrams, which resolve the internal stratigraphy at PIG for TWTs up to (1000-1200) ns. Additional information includes the considered averaging periods, density profiles, and type of annual dating markers for each study.

that the SMB decreases towards the interior so the contribution from this area to the total mass input will be less than that from the rest of the basin.

Additional SMB measurements were made with a ground penetrating radar during traverse T1 and published in Konrad et al. (2019). The authors selected the $\sim 1986$ reflection layer, which approximately coincides with the observed main reflector by ME14, and traced the layer along sections of the 900 km traverse, amounting to a total of a 613 km distance covered by GPR observations. The route of these observations closely follows the ASIRAS flight track and are both available at http://gis.istar.ac.uk/. Due to the limited maximum sampling depth of the ASIRAS and NP measurements, the 1985/86 reflection layer used by ME14 and Konrad et al. (2019) is not contained in most of our data. To benefit from the ASIRAS coverage while simultaneously accounting for its limited depth range, we manually traced the continuous 2005 reflection layer over a distance of 2367 km to derive mean annual SMB estimates for the 2005–2014 period. Due to the reported consistency between the GPR and airborne SMB measurements in Konrad et al. (2019), we limit the comparison of our results to the basin wide estimates by ME14. In addition, we assume that the effect of strain history, which could affect our SMB estimates at the fast flowing sections of PIG, is negligible. Konrad et al. (2019) conclude that the total effect over the whole catchment is small, even though it can have a very significant effect at some sites. However, this effect is expected to be further reduced for the shallower reflection layer depths from the ASIRAS measurements.

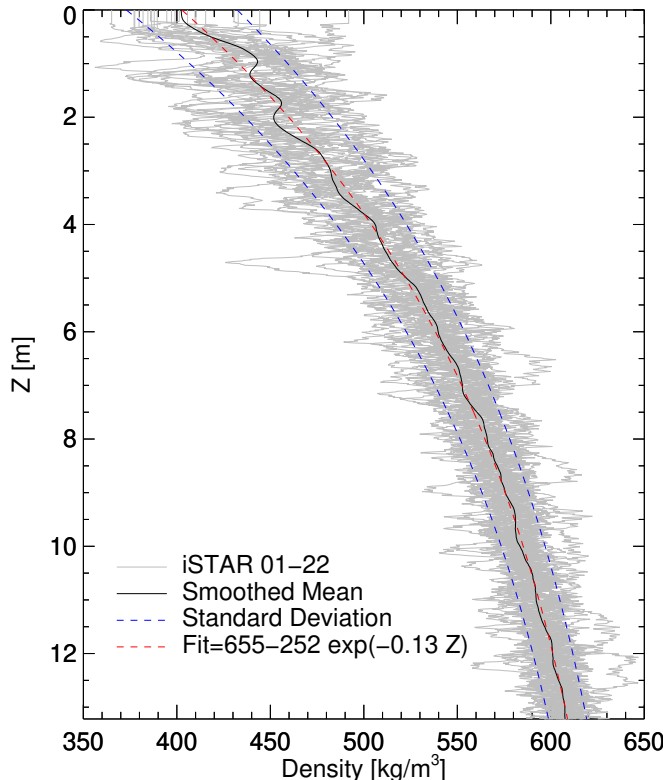

**Figure 2.** Compiled density–depth profiles from traverse T2 at all 22 iSTAR sites (grey lines). The black line denotes the smoothed mean profile, the red dashed line an exponential fit (units according to axis annotations), and the blue dashed lines the standard deviation intervals of the fit.

## 2.2 Neutron Probe measurements

NP measurements of snow and firn density were performed at all stations during both traverses as described in Morris et al. (2017). Further details on the calibration procedure, which is based on theoretical considerations, can be found in Morris (2008). A comparison with gravimetric density measurements at existing core profiles did not indicate a systematic bias between both measurement methods. To evaluate the effect of densification, the ground team repeated the density profiling in the same boreholes during traverse T2. Because the most recent accumulation is missing in these profiles, they drilled an additional borehole of less than 6 m depth and a nearby distance of about 1 m to capture it during traverse T2. The only exception is site 2, where the ground team decided to auger a completely new 14 m borehole for the density profiling due to poor data from the T1 hole.

The deep firn cores ($\sim 50$ m) shown in Fig. 1 were collected and analysed by the British Antarctic Survey. This analysis includes the annual variations with depth of the photochemical $H_2O_2$ tracer and density, which are phase shifted by about six months. According to Morris et al. (2017) the annual density variation is caused by alternating late austral summer/autumn

low-density hoar layer with winter snow which has densified under the influence of warm summer temperatures. The different processes, which modulate the density and $H_2O_2$ concentration with depth, allow for an independent determination of annual snow accumulation at the 10 deep core sites. No volcanic reference horizon was detected in the cores (R. Mulvaney, pers. comm.) and therefore limits the annual markers to the $H_2O_2$ and density profiles. Morris et al. (2017) applied an automatic annual layer identification routine to the vertical density profiles and used the annual $H_2O_2$ peak depths as an additional guidance for the annual layer dating. Thus, the depth–age scales from both annual markers are consistent.

We use a single regional density–depth profile we derive from the NP profiles of traverse T2 for the two-way-travel time (TWT) to depth conversion of the ASIRAS soundings. First, we merge the $\sim 13$ m and nearby $\sim 6$ m density–depth profiles at each site (except at site 2) by linearly relaxing their overlapping segments. To reduce the effect of lateral noise convolution, we limit the relaxation length to the overlapping segments that correlate well with each other. Then we align the intercepting depth–age scales to create a consistent depth–age scale for each compiled profile. The resulting 21 merged profiles and the single profile at site 2 are shown by the grey lines in Fig. 2. From these 22 profiles, we then determine a smoothed regional mean profile, which is denoted by the black line. Morris et al. (2017) observed a two-stage Herron and Langway (1980) type densification at PIG, with the stages separated by an additional transition zone. We achieve a good fit to our regional mean profile with a simple exponential function (red dashed line, Fig. 2), which we apply to the TWT to depth conversion. The blue dashed lines show the fitted standard deviation of the density as a function of depth. Following Medley et al. (2013), we consider the fitted standard deviation to be representative of the spatial uncertainty of the regional scale density–depth profile.

## 2.3 ASIRAS soundings

ASIRAS is a Ku-band radar altimeter which operates at a carrier frequency of 13.5 Ghz and a bandwidth of 1 GHz (Mavrocordatos et al., 2004). It was set to Low Altitude Mode (designed for heights less than 1500 m above ground) during its measurements at PIG. A Synthetic Aperture Radar (SAR)-processing of the collected data was performed, which yields the spatial resolution of the SAR level_1b data shown in Tab. 1. The associated cross track footprint is $\sim 15$ m. We use the electromagnetic wave speed $v = c/\sqrt{\epsilon'}$ to convert TWT to depth, where $c$ is the vacuum speed of light and $\epsilon'$ is the real part of the dielectric permittivity of the firn column. For the latter, we apply the commonly used empirical relation by Kovacs et al. (1995):

$$\epsilon'_{kov} = (1 + 0.845\rho_s)^2, \tag{1}$$

where $\rho_s = \rho/\rho_w$ is the specific gravity of snow (or firn) at current depth with respect to the water density $\rho_w = 1000 \, \mathrm{kg\,m^{-3}}$. An alternative model by Looyenga (1965) is

$$\epsilon'_{loo} = \left( \frac{\rho}{\rho_{ice}} \left[ \sqrt[3]{\epsilon'_{ice}} - 1 \right] + 1 \right)^3, \tag{2}$$

with $\epsilon'_{ice} = 3.17$ (Evans, 1965) and $\rho_{ice} = 917 \, \mathrm{kg\,m^{-3}}$. Sinisalo et al. (2013), who consider a similar depth range to this study, conclude that the difference between wave speeds based on Eq. (1) and (2) has a negligible impact on their SMB estimates. This is also the case for our estimates (see Sec. 3). The maximum depth of the radargrams is $\sim 30$ m based on the TWT to

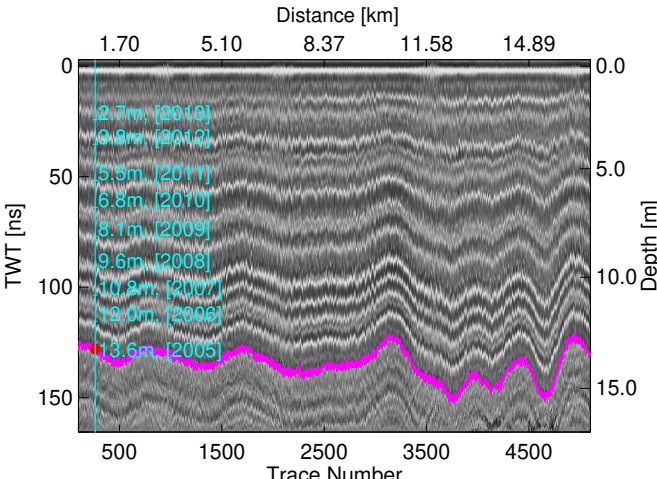

**Figure 3.** ASIRAS radargram at iSTAR site 21 with surface snow reflection centred at the origin of the TWT scale. Traced internal reflection layer highlighted in magenta, annual markers from NP profile at point of closest approach highlighted in cyan. The distance and trace numbers refer to the origin of the ASIRAS track segment 20156124 (see Tab. 2).

depth conversion from the fitted regional mean profile of density with depth and substituted Kovacs relation. The depth range
of resolved internal stratigraphy varies along the flight track, but the layering remains visible for most of the upper 13 m depth
covered by the NP measurements. Using the regional mean profile of density with depth, we determine the water equivalent
(w.e.) depth value for each waveform bin and calculate the mass per unit area between the selected reflection layer (magenta
line in Fig. 3) and surface. We assume that internal reflection layers are generated by the dielectric contrast at embedded thin ice
and hoar layers (Arcone et al., 2004, 2005) and that these layers are formed at regional scales around summer/autumn (Medley
et al., 2013). These layers may coincide with the annual density modulation, which we observe with the NP measurements.
Before the layer tracing, we apply an automatic-gain-control filter to all waveforms and limit their dynamic range to twice the
standard deviation centred around the mean amplitude of each waveform. This improved the signal contrast of the radargram.
Initially we tested a phase following algorithm of the Paradigm EPOS geophysical processing software to trace the selected
reflection layer semi-automatically. However, this method became unstable for lower contrast and cases with close layer spac-
ing. Furthermore, remaining SAR-processing artefacts were interfering with the phase following algorithm. Because of the
complex nature of the observed stratigraphy, as has been also reported by Konrad et al. (2019), manual layer tracing was used.
Following Richardson et al. (1997), we attempted to bridge distorted or merged layer segments whenever distinct characteris-
tics of a vertical layer sequence could be identified with confidence before and after the bridging. Different processes can lead
to distortion of the reflection layers, e.g. processes changing deposition of the annual snow layers or excessive rolling angles
of the airplane, while merging layers can result from low snow precipitation and ablative processes (e.g. wind-scouring) or
a combination of both. We checked the traced layer for possible mismatches which may have resulted from systematic errors in

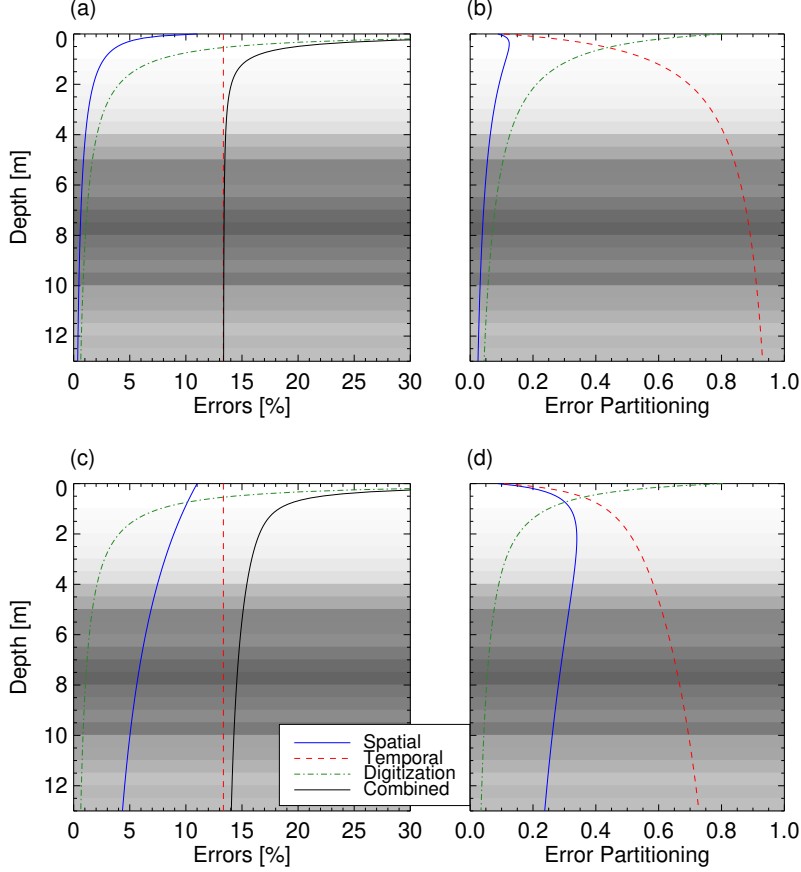

**Figure 4.** Spatial, temporal, digitization, and combined SMB measurement errors, which relate to the variability in density, dating uncertainty, and ASIRAS sampling accuracy, respectively: Relative errors (left panels) and error partitioning (right panels). Grey background shades indicate the depth distribution of the traced 2005 reflection layer with higher number concentrations towards darker shaded. (a,b) Based on error propagation according to Eq.(5). (c,d) Excluded spatial error cancellation in Eq.(5) [see main text] and considered for the final error estimation of this study.

the initial manual layer tracing at 34 cross-over points and 8 nearby flight track segments. Such mismatches were particularly observed along challenging profile sections and corrected by retracing the reflection layers, which yield the best match at the crossover points. In this sense, the layer tracing is performed independently from the annual layer dating at each iSTAR site.

## 2.4 Measurement error estimation

We attempt to trace the 2005 reflection layer, which is covered by all NP density–depth profiles. So far, we assumed that internal reflection layers form on an annual basis during summer/autumn but the potential formation of intra-annual reflection layers may challenge this assumption. For instance, Nicolas et al. (2017) found evidence of surface melt episodes over large parts

| Track Number | iSTAR Site | Latitude [deg] | Longitude [deg] | Elevation [m MSL] | Year | $\Delta D$ [m] | $N$ | Comment |
|---|---|---|---|---|---|---|---|---|
| 20156125 | 1 | -74.565 | -86.913 | 1362 | $2003.9 \pm 0.2$ | 75 | 51 | |
| 20156125 | 2 | -74.865 | -88.030 | 1195 | $(2009.88 \pm 0.03)$ | 8 | 5 | erroneous NP dating |
| 20156125 | 3 | -74.111 | -89.224 | 1032 | $2003.39 \pm 0.04$ | 11 | 7 | |
| 20156125 | 4 | -75.319 | -90.524 | 860 | $2004.9 \pm 0.1$ | 86 | 55 | |
| 20156110 | 4 | ⋮ | ⋮ | ⋮ | $2004 \pm 1$ | 811 | 422 | |
| 20156125 | 5 | -75.431 | -92.060 | 798 | $2006.4 \pm 0.1$ | 119 | 67 | |
| 20156125 | 6 | -75.456 | -93.718 | 708 | $2005.1 \pm 0.2$ | 177 | 121 | |
| 20156106 | 6 | ⋮ | ⋮ | ⋮ | $2006.0 \pm 0.6$ | 2000 | 856 | |
| 20156125 | 7 | -75.440 | -94.460 | 679 | missing | 226 | | noise |
| 20156115 | 7 | ⋮ | ⋮ | ⋮ | $2006.3 \pm 0.4$ | 304 | 251 | |
| 20156113 | 8 | -75.090 | -95.070 | 708 | $2004.8 \pm 0.2$ | 316 | 149 | |
| 20156109 | 8 | ⋮ | ⋮ | ⋮ | $2004.9 \pm 0.2$ | 470 | 277 | |
| 20156113 | 9 | -74.956 | -94.631 | 733 | $2003.1 \pm 0.1$ | 282 | 129 | |
| 20156113 | 10 | -74.442 | -93.448 | 867 | $2003.6 \pm 0.4$ | 9 | 5 | |
| 20156114 | 10 | ⋮ | ⋮ | ⋮ | $2004.9 \pm 0.1$ | 331 | 120 | |
| 20156114 | 11 | -74.620 | -92.700 | 909 | $2004.3 \pm 0.2$ | 121 | 37 | |
| 20156115 | 11 | ⋮ | ⋮ | ⋮ | $2004.3 \pm 0.2$ | 76 | 57 | |
| 20156115 | 12 | -74.998 | -93.930 | 762 | missing | 490 | | noise |
| 20156115 | 13 | -75.670 | -94.690 | 691 | $2005.5 \pm 0.2$ | 127 | 71 | |
| 20156102 | 13 | ⋮ | ⋮ | ⋮ | $2005.7 \pm 0.2$ | 56 | 35 | |
| 20156109 | 13 | ⋮ | ⋮ | ⋮ | $2004.7 \pm 0.1$ | 377 | 102 | |
| 20156107 | 14 | -75.805 | -94.231 | 749 | $2003.5 \pm 1$ | 325 | 171 | |
| 20156109 | 14 | ⋮ | ⋮ | ⋮ | $2005.0 \pm 0.6$ | 350 | 174 | |
| 20156107 | 15 | -75.750 | -96.730 | 712 | $2005.15 \pm 0.09$ | 485 | 257 | |
| 20156120 | 15 | ⋮ | ⋮ | ⋮ | $2005.92 \pm 0.2$ | 190 | 112 | |
| 20156107 | 16 | -75.926 | -96.898 | 763 | $2004.19 \pm 0.05$ | 413 | 246 | |
| 20156103 | 16 | ⋮ | ⋮ | ⋮ | $2003.8 \pm 0.1$ | 314 | 181 | |
| 20156120 | 17 | -75.740 | -97.930 | 716 | missing | 60 | | noise |
| 20156121 | 18 | -75.617 | -99.073 | 527 | $2004.33 \pm 0.09$ | 186 | 98 | |
| 20156120 | 18 | ⋮ | ⋮ | ⋮ | $(2003.2 \pm 0.2)$ | 297 | 143 | extrapolated |
| 20156126 | 18 | ⋮ | ⋮ | ⋮ | $(2002.0 \pm 0.4)$ | 867 | 245 | extrapolated |
| 20156121 | 19 | -75.803 | -99.048 | 704 | $(1996.8 \pm 0.3)$ | 230 | 44 | extrapolated |
| 20156122 | 20 | -76.404 | -99.828 | 1096 | $2005.12 \pm 0.06$ | 15 | 2 | |
| 20156122 | 21 | -76.224 | -100.770 | 1075 | $2005.73 \pm 0.02$ | 115 | 40 | |
| 20156124 | 21 | ⋮ | ⋮ | ⋮ | $2005.71 \pm 0.07$ | 26 | 15 | |
| 20156124 | 22 | -75.804 | -100.280 | 819 | $2005.65 \pm 0.06$ | 21 | 13 | |

**Table 2.** Dated reflection layer year at nearby iSTAR sites. "Track number" refers to the ASIRAS flight track naming convention (year of measurement season [4 digits], measurement type [1 digit], profile segment number [3 digits]), "$\Delta D$" is the closest distance between the ASIRAS track and iSTAR site, and $N$ is the number of picking samples considered for layer dating. Years in brackets are discarded from the regional layer age estimation as follows: Significant departure between traced layer and depth–age scale at site 2 (see main text), layer gaps due to high *noise* levels in the radargram, layer is significantly exceeding the dated NP profile depth (values in brackets indicate *extrapolated* depth–age values).

of WAIS in response to warm air intrusion events. Scott et al. (2010) observed a strong reflection layer, which coincides with an exceptional melt layer at 22 m depth at one PIG ice core location. These findings suggest that intra-annual reflection layers can form at the basin scale, even though, the formation is less frequent as it appears to be related to the complex coupling between different atmospheric modes (e.g. Nicolas et al., 2017; Donat-Magnin et al., 2020). The frequency of intra-annual reflection layer formation may change towards the coast, where the snow accumulation is high. For instance, Fig. 3 shows

additional reflection layers with respect to the annual density markers from the NP measurements at site 21. Extreme solid precipitation events may also impact the density modulation with depth (Turner et al., 2019), which is considered for the depth–age scale based on the NP measurements. Snow erosion may remove annual markers where accumulation rates are low. In addition to annual layer counting errors, the timing between the reflection layer formation and snow densification may be offset during summer/autumn. All these factors challenge the tracing and dating of the 2005 reflection layer but combining

the stratigraphic information from the ASIRAS and iSTAR observations helps reducing the risk of systematic errors from erroneous layer counting. To account for the remaining risk in terms of isochronal accuracy, we assign an annual layer tracing uncertainty of $\overline{\delta t} = \pm 1$ years.

     Following Morris et al. (2017), we define mass balance years between the density peaks in the NP profiles (nominally 1st of July). For instance, the mass balance year 2013 begins at the second annual density peak below the surface (nominally

01 July 2013) and ends at the first peak (01 July 2014). Based on annual density markers, we can relate the snow and firn depth at each iSTAR site to its associated age and determine the reflection layer age from its depth at each cross section. Here, we use an exponential fit of the local density–depth profile for the TWT to depth conversion. The lateral displacement $\Delta D$ between the point of closest approach of the flight track and iSTAR site adds to the reflection layer dating uncertainty. We therefore consider all $N$ points which lie within a $2\Delta D$ interval along the flight track and which is centred at the point

of closest approach for the layer dating. Based on the local depth–age scale, we relate the estimated depths of $N$ points to their ages and assign the final layer date to their mean value. To account for the mass balance year definition above, we add six months to the mean layer date, which is listed for all iSTAR positions in Tab. 2. In addition, we estimate the dating uncertainty from the $N$ lateral estimates by their standard deviation $\sigma_x$. In this sense, our error estimate is more conservative than the standard error of the mean. Furthermore, we assume that the uncertainty due to local variation in the stratigraphy is

isotropic, which does not generally need to be true. However, according to Tab. 2 the overall impact of this effect is one order of magnitude smaller than the variability of layer age values among all iSTAR sites in most cases. As indicated in Tab. 2, we excluded dating estimates around iSTAR sites 2 and 19. In both cases, our layer tracing revealed a large offset contrary to the neighbouring iSTAR sites. Possible reasons for these offsets could be systematic errors in the layer dating from the NP profiles, the variability of internal stratigraphy between the ASIRAS measurements and their closest approach to both iSTAR sites or

systematic errors in the manual reflection layer tracing. The remaining exclusion of layer age estimates at iSTAR sites 7, 12, and 18 is either due to high noise levels of the radargram or reflection layer depths significantly exceeding the NP depth–age scales. Following Konrad et al. (2019) we estimate the final reflection layer year by the mean of dating values at each site with an uncertainty of $\Delta t = \sqrt{\overline{\delta t}^2 + \delta \overline{t}^2 + \overline{t}_x^2}$, with the standard deviation of dating estimates $\delta \overline{t}$ and the propagated error $\overline{t}_x = 1/n \sqrt{\sum_i^n \sigma_x(i)^2}$ from the $n$ lateral error estimates around each iSTAR site ($i$ = site number), which we introduced in

addition. The resulting reflection layer dating estimate is $T = 2004.8 \pm 1.4$, which corresponds to a layer age of $a = 10.1 \pm 1.4$. The associated average surface accumulation rate $\dot{b}$ in terms of w.e. depth per year is

$$\dot{b} = \frac{1}{a\rho_w} \sum_{i=1}^{m} \delta z_i \rho_i, \tag{3}$$

where $\delta z_i$ is the $i^{\text{th}}$ depth increment of the radar waveform and $\rho_i$ is the associated density. Substitution of the wave propagation speed for $\delta z_i$ yields

$$\dot{b} = \frac{1}{a\rho_w} \sum_{i=1}^{m} \frac{c t_s}{\sqrt{\epsilon_i'}} \rho_i, \tag{4}$$

where $t_s = 0.37$ ns is the ASIRAS vertical bin sampling time (i.e. $0.5 \times$ TWT per bin), and $\epsilon_i'$ refers to the permittivity value at the $i^{\text{th}}$ bin. To avoid any confusion with previous summations, the final index $m$ refers to the traced waveform bin at the reflection layer depth. It is evident from Eq. (4) that the spatial uncertainty of the density profile affects both the integration depth and incremental mass. Medley et al. (2013) and ME14 estimated the spatial uncertainty from the resulting SMB change

by directly applying the standard deviation fits of their regional density profiles to the TWT to SMB conversion. Instead, we may propagate the error in Eq. (4), assuming that errors are uncorrelated and normally distributed. Based on the Kovacs relation according to Eq. (1) we account for the temporal, spatial, and digitization:

$$\Delta\dot{b} = \frac{c t_s}{a\rho_w} \sqrt{\underbrace{\sum_{i=1}^{m} \left(\frac{\Delta\rho_i}{\epsilon_{kov,i}'}\right)^2}_{\text{spatial}} + \underbrace{\left(\frac{\Delta a}{a} \sum_{j=1}^{m} \frac{\rho_j}{\sqrt{\epsilon_{kov,j}'}}\right)^2}_{\text{temporal}} + \underbrace{\left(\frac{1}{3} \sum_{k=m-1}^{m+1} \frac{\rho_k}{\sqrt{\epsilon_{kov,k}'}}\right)^2}_{\text{digitization}}}, \tag{5}$$

where $\Delta a = \pm 1.4$ years is the temporal uncertainty and $\Delta\rho_i$ are the standard deviation intervals according to Fig. 2. Due to

195 the small incremental density change of $< 0.7\%$ along the entire profile, we approximate the digitization error by the mean SMB value of three consecutive bins centred at the final profile bin of the current integration depth. Figure 4 (a) displays the propagated individual measurement error components as well as the combined measurement error according to Eq. (5) as a function of geometric depth. In addition, we include the error partitioning in (b). The grey background shades highlight the distribution of layer depths to visualise the relevant error range of our SMB estimates, which peaks around (5, 8, and 10) m

(darker shades). In comparison with Medley et al. (2013) and ME14, we find that our spatial error estimate based on Eq. (5) is reduced by about one order of magnitude while the standard deviation fits of their regional density profiles cover a similar range compared to ours. We may ignore the spatial error compensation in Eq. (5) by replacing the root-sum-of-squares (RSS) with absolute values: $\sum_{i=1}^{m} \left(\Delta\rho_i/\epsilon_{kov,i}'\right)^2 \rightarrow \left(\sum_{i=1}^{m} |\Delta\rho_i/\epsilon_{kov,i}'|\right)^2$. Hence, to comply with the studies above, we consider the more conservative spatial error propagation based on the sum of absolute values, but we keep the RSS of individual error

components for the combined measurement error estimate as shown in Fig. 4 (c-d). Following these assumptions, we find that our measurement error estimate is still dominated by the temporal layer dating uncertainty for most of the traced layer depths, but the spatial error reaches a similar range to that reported in Medley et al. (2013). We consider the combined measurement error based on Fig. 4 (c-d) for the SMB estimates of this study.

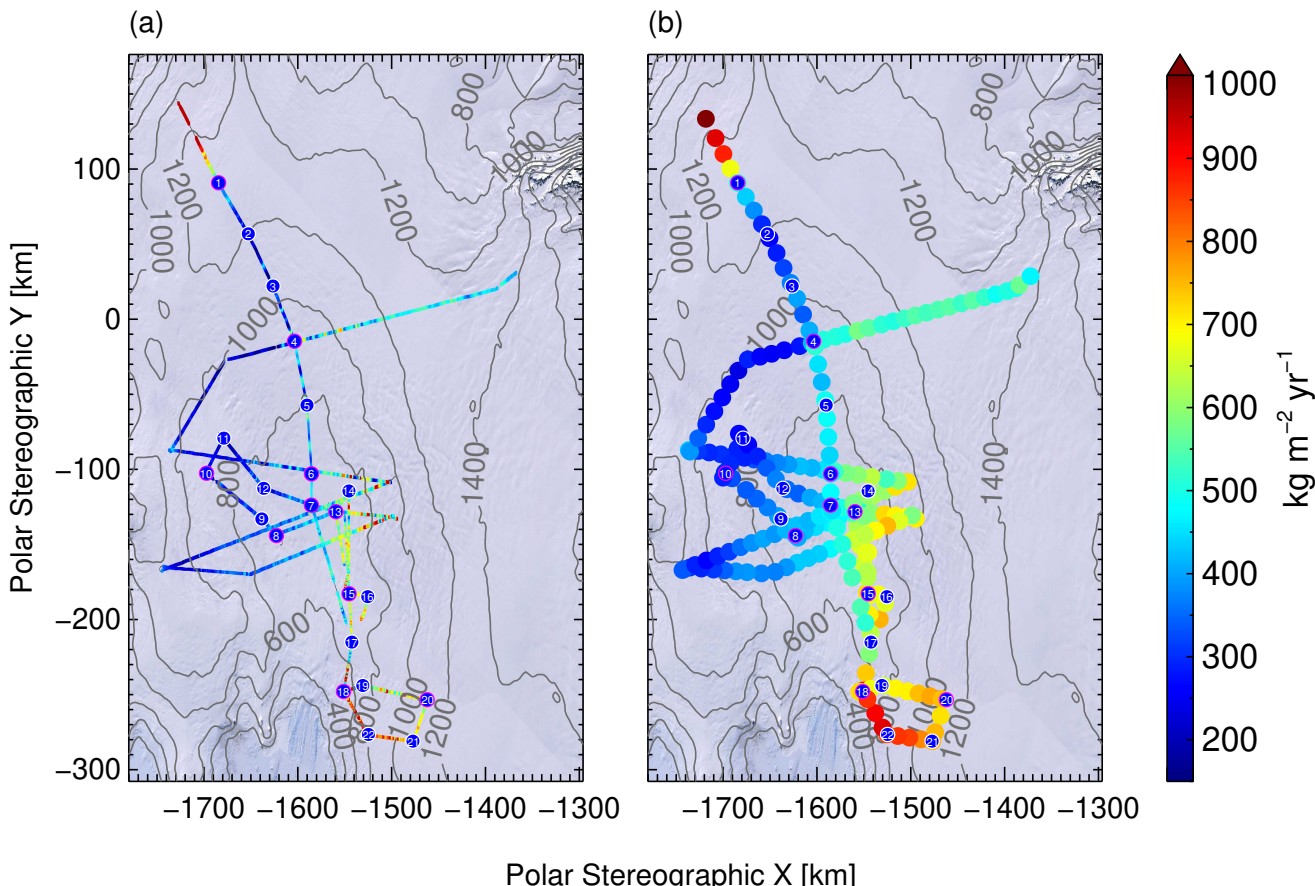

**Figure 5.** Traced annual mean SMB between November 2004 and December 2014 from ASIRAS soundings with overlayed contour lines from a digital elevation model (Helm et al., 2014) and Landsat background imagery (U.S., Geological Survey, 2007). Circles with numbers denote the iSTAR sites. (a) High spatial SMB resolution, (b) smoothed and downsampled SMB estimates.

### 2.5 Kriging scheme

We focus on the regional scale variability of the SMB distribution at PIG. Figure 5 shows our high resolution (i.e. metre-scale) SMB estimates as well as smoothed SMB values with contour lines from a digital elevation model (DEM) by Helm et al. (2014). We use the same 25 km along-track smoothing window as ME14 and choose a sampling interval of half the smoothing window length. We initially tested the same interpolation scheme as described in ME14 to estimate a regional scale SMB field for the PIG basin from our smoothed SMB points. This scheme is based on the ordinary kriging (OK) algorithm, a widely used

geostatistical interpolation technique (e.g. Isaaks and Srivastava, 1991). Instead of a direct OK interpolation of smoothed SMB observations, ME14 consider the residual SMB values with regard to an ordinary least squares linear regression model for the Thwaites–PIG basin area with northing, easting, and elevation as explanatory variables. This, in turn, yields a small degree of

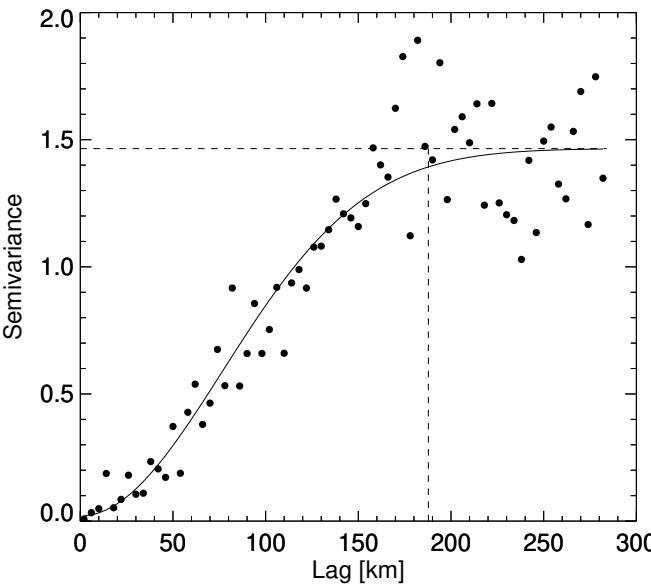

**Figure 6.** Experimental semivariogram of log-transformed smoothed SMB observations (dots), Gaussian fit model (black solid line), sill and practical range parameter (dashed lines).

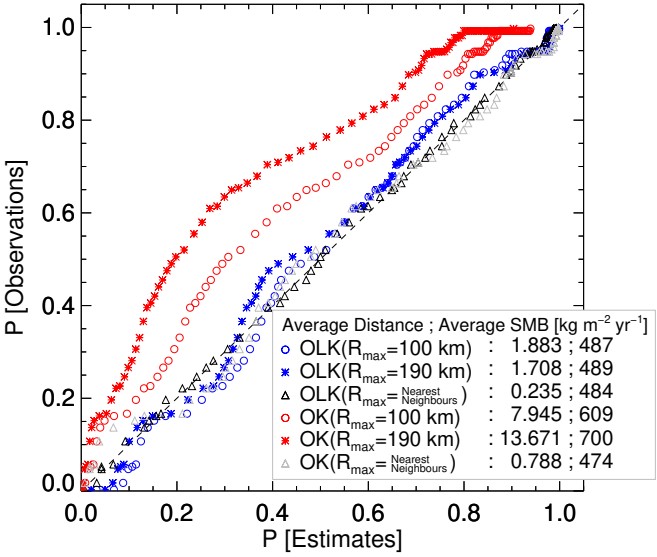

**Figure 7.** PP-plots between SMB observations and estimates based on OK and OLK interpolation methods for varying thresholds of their maximum distance $R_{max}$. The dashed 1:1 line indicates the complete PP-agreement between observations and estimates. Average PP-distances (see main text for definition) and SMB values are shown in the legend.

skewness $< 0.5$ with respect to the residual SMB distribution. However, we failed to reduce the skewness of residual SMB values from our estimates effectively by the same method, which may be due to the different aerial coverage considered in our

regression model. Examination of the DEM contour lines in Fig. 5 reveals that a simple relation between surface elevation and SMB is not evident, which may hint that the prevailing synoptic scale weather conditions at the Amundsen and Bellingshausen Sea sectors in combination with the precipitation shadowing effect of the Eights Coast mountain range (Fig. 1) require a more sophisticated model to capture the SMB at the PIG basin scale. We therefore searched for an alternative approach to generate krige estimates from the SMB sample population of this study without the use of a regression model. Such alternative, which

is also mentioned in ME14, is a logarithmic transformation of the SMB observations prior to the OK interpolation:

$$\dot{B}(x_0) = \ln\left(\dot{b}(x_0) + C\right),\tag{6}$$

where $C$ is an arbitrary constant and $x_0$ represents the current interpolation location. After the OK interpolation of transformed SMB observations, the estimates must be transformed back into the original measurement scale. This backtransformation requires the addition of a nonbias term for each OK estimate to ensure that the expected value is equal to the sample mean and

230 that the smoothing effect is adequately compensated (i.e. resulting estimates reproduce the sample histogram and sample mean [Yamamoto, 2007]). We implemented such ordinary logarithmic kriging (OLK) method in our analysis by adopting the 4-step post-processing algorithm proposed by Yamamoto (2007) for the estimation of nonbias terms. According to Yamamoto (2008), OLK does not necessarily require a log-normal sample distribution to produce improved estimates in terms of local accuracy. Furthermore, Yamamoto (2007) tested the impact of constant $C$ according to Eq. (6) and found that a data translation towards

higher values yields an approximation from OLK to OK estimates, thus, eliminating the advantage of improved sample mean reproduction and local accuracy of OLK estimates. Indeed, we find that adding a negative constant $C$ to all SMB values, such that the lowest SMB value reaches $0.1 \ \mathrm{kg\,m^{-2}yr^{-1}}$, yields an improved reproduction of the observation data characteristics. Figure 6 shows the experimental isotropic semivariogram of our log-transformed SMB observations from Fig. 5 (b) together with a Gaussian model fit with a practical range of $\sim 190 \ \mathrm{km}$, i.e. the range at which the spatial autocorrelation of sample

points is vanishing. Following Yamamoto (2005, 2007), we investigate the reproduction of observational data characteristics by means of PP-plots (i.e. percentiles of cumulative distributions of observations and estimates against each other). Figure 7 shows the PP-plots for our OLK and OK interpolation constrained to a maximum estimation range threshold $R_{\mathrm{max}}$ with regard to the closest ASIRAS measurement locations of $100 \ \mathrm{km}$ and $190 \ \mathrm{km}$, and nearest neighbour locations. By comparison with Fig. 6, the $100 \ \mathrm{km}$ and $190 \ \mathrm{km}$ distances (dashed lines) approximately correspond to the lag distances at which the semi-

variogram has reached half the sill and where it has levelled off, respectively. In addition, the average distance of PP-points from the 1:1 line according to the definition in Yamamoto (2005) as well as the average SMB values for the OK and OLK estimates are shown in the legend. Both, the nearest neighbour OK and OLK average SMB estimates are close to the average SMB observation value of $474 \ \mathrm{kg\,m^{-2}yr^{-1}}$. However, after increasing the range threshold $R_{\mathrm{max}}$ to $100 \ \mathrm{km}$ and $190 \ \mathrm{km}$, it is evident from Fig. 7 that the best match exists between the observation and OLK estimation values. Hence, we limit our analysis

to these values in the following.

Aside from the choice of the translational constant $C$ and semivariogram model, we choose the method proposed by Deutsch (1996) to correct for negative kriging weights (Yamamoto, 2000) and constrain all processing steps of the OLK estimation to the 16 nearest neighbours for each estimate according to the quadrant criterion. Depending on the neighbourhood considered

the effect of smoothing as well as local stationarity of observation data is affected. As a guidance for our final setting, we aimed at generating an optimal PP-relation according to Fig. 7, but also considered potential artefacts, which may arise from the OLK procedure.

In addition to each OLK estimate, we calculate the associated interpolation error. While ME14 choose the kriging standard deviation as a measure of interpolation error, our error estimation is based on the interpolation standard deviation $S_0$ introduced by Yamamoto (2000) for two reasons. Firstly, as shown by the author, $S_0$ represents a more complete measure of local accuracy and has, therefore, been implemented in the post-processing algorithm in Yamamoto (2007). Secondly, for the OLK method we need a corresponding backtransformation of the interpolation error from the logarithmic to the measurement scale, which has been investigated for $S_0$ in Yamamoto (2008). Thus, we adopted the proposed backtransformation of $S_0$ in this study.

Following ME14, we estimate the total error of each SMB estimate by the RSS of the measurement error and backtransformed $S_0$. The measurement error is estimated by generating 500 realisations of OLK SMB estimates with added noise to the smoothed SMB observations, which follows a normal distribution with a mean of zero and standard deviation equal to the measurement error of the SMB observation at $x_0$.

We have to keep in mind that the basin wide SMB OLK estimation is limited in terms of the practical range according to Fig. 6. By comparison with the flight track shown in Fig. 1, even when considering the practical range as a maximum threshold for the spatial SMB estimation, we do not cover the entire PIG basin (see Fig. 8). Hence, for the calculation of total mass input to the PIG basin, we replace SMB OLK estimates with modelled SMB from a regional climate model at distances where the spatial autocorrelation of measurements is low. In the next section, we either consider SMB estimates from the RACMO2.3p2 (van Wessem et al., 2018) regional climate model (in the following abbreviated as RACMO) and the Modèle Atmosphéric Régional (MAR) according to Donat-Magnin et al. (2020).

## 3    Results

### 3.1    Regional scale SMB distribution

Based on the adopted OLK interpolation scheme, we produced the mean annual SMB map for the PIG basin from the ASIRAS observations in Fig. 8 (a). SMB observations and estimates are colour coded with the same scale. Each estimate covers a pixel size of $\sim 5$ by $5\ \mathrm{km}^2$ and refers to the averaging period between November 2004 and December 2014. The two surrounding dashed lines indicate the $100\ \mathrm{km}$ and $190\ \mathrm{km}$ maximum distances from the ASIRAS measurement point cloud discussed earlier. The red triangle denotes an artificial interpolation cluster of 8 pixels with SMB values greater than $2000\ \mathrm{kg\,m}^{-2}\mathrm{yr}^{-1}$, which we discuss in Sec. 4.4. Furthermore, some streak artefacts are visible from the interpolation, which are mainly caused by the quadrant criterion of the OLK estimation. Increasing the number of nearest neighbours helps reducing these artefacts but at the cost of PP-agreement in terms of Fig. 7. We therefore kept the OLK settings according to Fig. 8 (a) hereinafter.

Panels (c,d) show mean annual SMB estimates for the same period based on RACMO and MAR simulations, respectively. The horizontal resolution of simulated SMB is $27\ \mathrm{km}$ for RACMO and $10\ \mathrm{km}$ for MAR runs. ASIRAS and model based estimates show similar main characteristics, i.e. increasing SMB rates towards the Amundsen Sea coastline, decreasing SMB

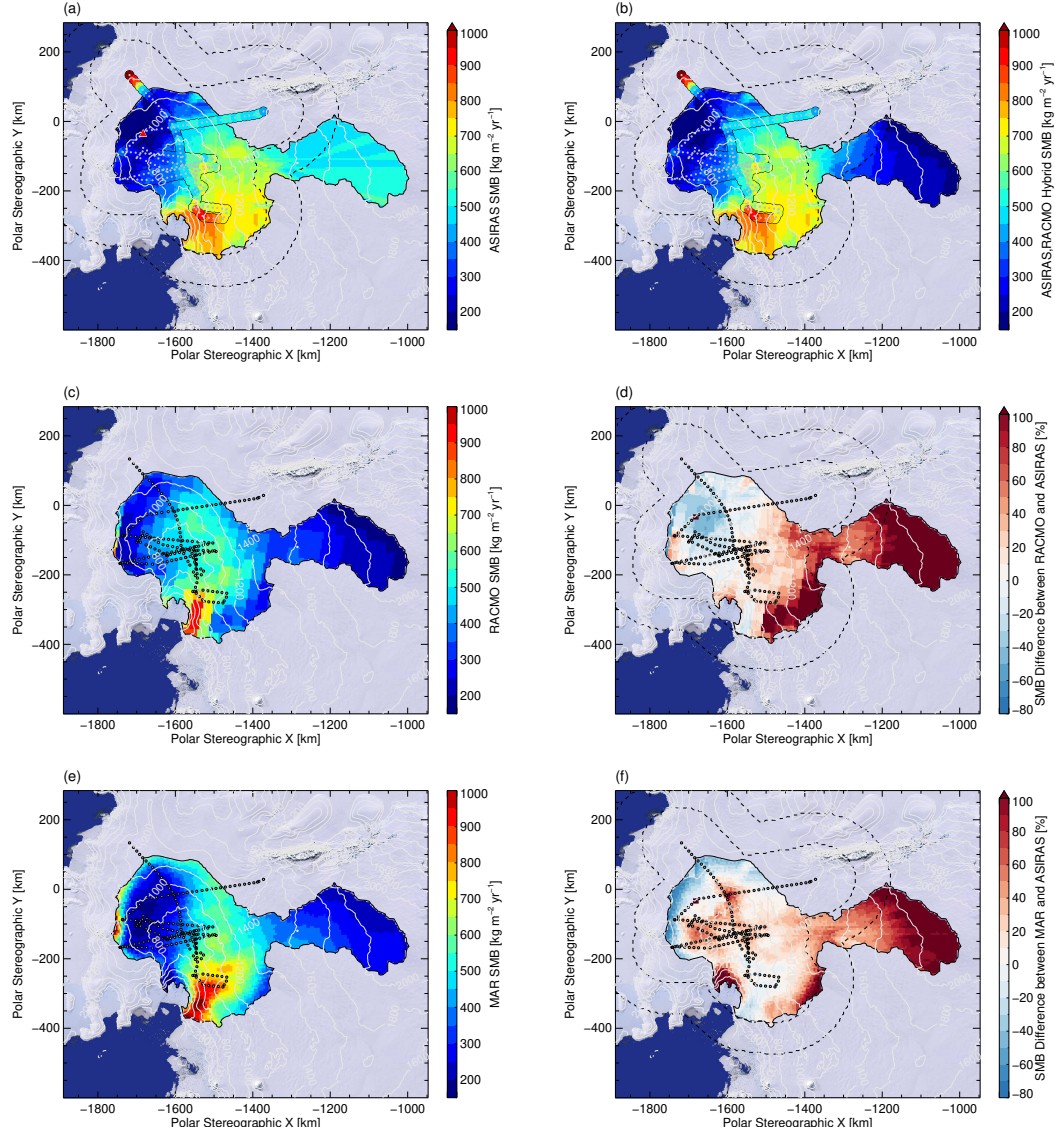

**Figure 8.** (a) ASIRAS annual mean SMB OLK estimates between November 2004 and December 2014. Measurement points colour coded with the same SMB scale. Red triangle denotes the position of an interpolation artefact (see Sec. 4.5). (b) Hybrid SMB map with ASIRAS estimates (a) linearly relaxing into RACMO SMB estimates (c), which have been extracted for the same averaging period. Dashed lines denote the 100 km and 190 km maximum distance to the ASIRAS estimates and confine the transition zone between ASIRAS and model based SMB estimates (a,b,d,f). (d) Relative SMB change from RACMO to ASIRAS, i.e. $[(a) - (c)]/(c) \times 100$. (e,f) MAR SMB fields and relative difference to ASIRAS by analogy with (d). Background imagery taken from U.S., Geological Survey (2007) for all panels.

rates towards the inland, and a region of low SMB in response to the shadowing effect from the Eights Coast mountain range. Similar characteristics also exist for the SMB map generated by M14. Furthermore, the ASIRAS observations start to capture

the transition to higher snow accumulation at the ice divide along the Eights Coast mountain range, as indicated by both regional climate models. This is best seen in the high resolution observations north of iSTAR site 10 according to Fig. 5.

Figure 8 (d,f) shows the relative difference between model and ASIRAS SMB estimates as defined in the caption. Local variations can be found in the agreement between ASIRAS and model based estimates. Among others, the shadowing effect appears to be more pronounced in the MAR and ASIRAS estimates than in the RACMO estimates. Furthermore, the MAR estimates tend to be lower at the central flight lines compared to the RACMO and ASIRAS estimates, whereas the agreement is the best between the MAR and ASIRAS estimates near coastal iSTAR sites. A common feature of the ASIRAS estimates is the much less pronounced SMB gradient towards the southern interior compared to both model estimates but also to the M14 estimates. This can be explained by the missing observational constraints in this region. We therefore generated hybrid SMB maps where ASIRAS estimates linearly relax into either MAR or RACMO estimates between the 100 km to 190 km range interval (dashed lines), as shown in Fig. 8 (b) for complementary ASIRAS–RACMO estimates. The range interval was selected based on the spatial autocorrelation in terms of Fig.6. It is evident from Fig. 8 (d,f) that the SMB gradient towards the southern interior is not the same for the MAR and RACMO simulations. Hence, the selection of complementary model data will impact total mass input estimates for the PIG basin.

## 3.2 Total mass input

Spatial integration of annual mean SMB from our generated hybrid maps yields the total mass input for the PIG basin, which we denote by $\Sigma_+$. Table 3 summarizes $\Sigma_+$ and further statistical SMB characteristics for different data sets and basin definitions according to Fig. 9 (d). Here, we replaced the interpolation artefact highlighted in Fig. 8 (a) with averaged values from neighbouring pixels. $\Sigma_+$ uncertainty estimates refer to the RSS of the interpolation and measurement error-grids (Fig. 9(c) in accordance with ME14. Because of a missing error-grid for simulated SMB, we consider the total combined error for the entire PIG basin as a conservative error estimate. In this sense, we are augmenting the missing model error estimation. To quantify the relative contribution of ASIRAS to the hybrid SMB estimates, OLK area and OLK $\Sigma_+$ denote the relative contribution in terms of covered land area and integrated SMB, respectively. For comparison with the hybrid based estimates of this study, we include results from RACMO, MAR, and ME14, which we converted from w.e. depth to SI units. Because of the different averaging periods between this study and ME14, we added model estimates in brackets, which we extracted based on the same averaging period as for the ME14 results.

**Pine Island and Wedge Zone**

The Pine Island $\Sigma_+$ values are in agreement between all data sets within the estimated error margins. This is different for the Wedge area, where the RACMO $\Sigma_+$ estimates are between 35–40 % lower compared to the estimates of this study and ME14. Increasing the averaging time of RACMO estimates to the 1985–2009/10 period of the ME14 results yields an increase of $\Sigma_+$ by 2% for the Pine Island and 8% for the Wedge area. However, the RACMO based total mass input to the Wedge area remains below the observational error margins. In comparison with RACMO and MAR estimates, we find that MAR based $\Sigma_+$ values are about 5% higher for Pine Island and 38% higher for the Wedge area. The higher MAR SMB compared to RACMO towards

| | | Gt yr$^{-1}$ | kg m$^{-2}$yr$^{-1}$ | | | | % | |
|---|---|---|---|---|---|---|---|---|
| Data Set | Basin | $\Sigma_+$ | $\mu_{\dot{b}}$ | $\sigma_{\dot{b}}$ | min | max | OLK Area | OLK $\Sigma_+$ |
| | Pine Island | $69.0 \pm 17.6$ | 421 | 195 | 147 | 958 | 72.1 | 82.9 |
| | Wedge | $10.9 \pm 1.6$ | 729 | 48 | 591 | 847 | 99.5 | 99.6 |
| Hybrid | Pine Island and Wedge | $79.9 \pm 19.2$ | 447 | 206 | 147 | 958 | 74.4 | 85.2 |
| ASIRAS, RACMO | PIG (Mouginot et al., 2017) | $77.5 \pm 19.2$ | 439 | 198 | 132 | 958 | 75.3 | 85.1 |
| | PIG (Zwally et al., 2012) | $92.4 \pm 22.4$ | 443 | 195 | 79 | 958 | 74.6 | 84.0 |
| | Pine Island | $71.2 \pm 17.6$ | 434 | 185 | 188 | 958 | 72.1 | 80.5 |
| | Wedge | $10.9 \pm 1.6$ | 729 | 48 | 599 | 847 | 99.5 | 99.6 |
| Hybrid | Pine Island and Wedge | $82.1 \pm 19.2$ | 459 | 195 | 188 | 958 | 74.4 | 83.1 |
| ASIRAS, MAR | PIG (Mouginot et al., 2017) | $79.5 \pm 19.2$ | 451 | 188 | 188 | 958 | 75.3 | 83.2 |
| | PIG (Zwally et al., 2012) | $94.7 \pm 22.4$ | 455 | 188 | 0 | 1258 | 74.6 | 81.9 |
| | Pine Island | $64.7 ; 65.8$ | $395 ; 401$ | $147 ; 152$ | $147 ; 144$ | $935 ; 989$ | | |
| RACMO | Wedge | $6.6 ; 7.1$ | $439 ; 476$ | $180 ; 189$ | $287 ; 315$ | $935 ; 989$ | | |
| (Nov.2004–Dec.2014 ; | Pine Island and Wedge | $71.2 ; 72.9$ | $398 ; 408$ | $151 ; 157$ | $147 ; 144$ | $935 ; 989$ | | |
| Jul.1985–Jan.2010) | PIG (Mouginot et al., 2017) | $70.6 ; 72.1$ | $400 ; 409$ | $150 ; 157$ | $132 ; 131$ | $935 ; 989$ | | |
| | PIG (Zwally et al., 2012) | $81.6 ; 83.6$ | $392 ; 401$ | $151 ; 158$ | $0 ; 0$ | $935 ; 1005$ | | |
| | Pine Island | $67.7 ; 69.2$ | $413 ; 422$ | $173 ; 176$ | $86 ; 126$ | $1176 ; 1162$ | | |
| MAR | Wedge | $9.1 ; 9.3$ | $607 ; 622$ | $195 ; 201$ | $336 ; 350$ | $1131 ; 1162$ | | |
| (Nov.2004–Dec.2014 ; | Pine Island and Wedge | $76.8 ; 78.5$ | $430 ; 439$ | $183 ; 186$ | $86 ; 126$ | $1176 ; 1162$ | | |
| Jul.1985–Jan.2010) | PIG (Mouginot et al., 2017) | $74.4 ; 76.1$ | $422 ; 431$ | $184 ; 187$ | $16 ; 48$ | $1176 ; 1162$ | | |
| | PIG (Zwally et al., 2012) | $87.0 ; 89.1$ | $417 ; 428$ | $187 ; 189$ | $-53 ; -36$ | $1802 ; 1771$ | | |
| M14 | Pine Island | $67.3 \pm 6.1$ | 400 | 130 | 210 | 840 | | |
| | Wedge | $11.0 \pm 0.7$ | 590 | 160 | 160 | 330 | | |

**Table 3.** Spatially integrated SMB ($\Sigma_+$), mean ($\mu_{\dot{b}}$), standard deviation ($\sigma_{\dot{b}}$), minimum, and maximum SMB based on different data sets and basin definitions according to Fig. 9 (d). For the hybrid SMB estimates of this study, the areal contribution as well as the contribution of spatially integrated SMB from the ASIRAS estimates is denoted by OLK Area and OLK $\Sigma_+$ respectability. In addition to the November 2004 to December 2014 averaging period of the hybrid estimates, RACMO and MAR estimates separated by semicolon refer to the July 1985 to January 2010 averaging period in accordance with the results from M14.

the southern interior yields a 3% increase for hybrid SMB estimates based on complementary MAR estimates. Considering the additional SMB properties according to Tab. 3, the hybrid based SMB estimates of this study show the largest variability, except for the Wedge area.

### Additional basin definitions

Table 3 includes results based on two additional basin definitions for PIG. Figure 9 (d) shows a composite plot of all basin definitions used here. The surface areas range between 176.5, 178.6, and $208.8 \times 10^3$ km$^2$ for the PIG basin (including Wedge) according to the definitions of Mouginot et al. (2017), Fretwell et al. (2013), and Zwally et al. (2012). With regard to the basin definition according to Mouginot et al. (2017), $\Sigma_+$ increases by about 3 % for the definition by Fretwell et al. (2013) and between 15 % to 19 % for the definition by Zwally et al. (2012) depending on which data set is considered according to Tab 3.

## 4  Discussion

We discuss first the pronounced differences between annual layer dating from ASIRAS reflection and neutron probe density profiles at some sites and then secondly the systematic differences in SMB distribution between the results of this study and those of ME14, RACMO, and MAR.

### 4.1  Local SMB departures

Key to the evaluation of our selected internal reflection layer is its isochronic nature, which we assume based on matched depth–age relations from the iSTAR ground truthing measurements. One may argue that these measurements can be subject to local noise in the density profile, which would challenge any comparison with nearby radar observations. For instance, Laepple et al. (2016) observed dominating stratigraphic noise at single pit density profiles near Kohnen station (East Antarctic plateau, Dronning Maud Land). Stacking of multiple profiles is one possibility to filter out noise. While this is not possible for the single iSTAR sites, the estimated dating uncertainty of $\pm 1.4$ years according to this study suggests that iSTAR ground truthing measurements at PIG are less prone to stratigraphic noise, which is most likely to be related to the higher SMB compared to $\sim 70$ kg m$^{-2}$yr$^{-1}$ nearby Kohnen station (Laepple et al., 2016). However, on a few occasions we identified larger departures in the annual layer dating, as it is the case for iSTAR site 2 (Tab. 2). While the layer tracing appears to be in agreement between site 1 and 3, the annual layer dating at site 2 would suggest an SMB of $\sim 290$ kg m$^{-2}$yr$^{-1}$ at the traced layer cross section rather than $\sim 150$ kg m$^{-2}$yr$^{-1}$ based on the 2004.8 layer dating of this study. Accordingly, local SMB results would increase by $\sim 100\%$, if we used the uncorrected depth–age scale at site 2, which most likely indicates a systematic error in the measurement scale. This is further corroborated by the measured SMB of $140$ kg m$^{-2}$yr$^{-1}$ at site 2 for the most recent 2014 layer, but also measured density and strainrate profiles suggest a mean annual SMB of $200$ kg m$^{-2}$yr$^{-1}$ based on the Herron and Langway (1980) stage 1 equation, which are both in a better agreement with the collocated ASIRAS based results. In this sense, the ASIRAS results allow us to be more confident of the site 2 strainrate measurements and therefore add to the densification analysis of Morris et al. (2017). The local SMB estimates near site 2 from ME14 and RACMO are within the 200

to $300\,\mathrm{kg\,m^{-2}yr^{-1}}$ range, but lack the local precision of ASIRAS measurements and therefore could not explain the measured density and strainrate profiles at site 2. The bias to the ASIRAS observations also exists for the MAR estimates, which reach the $350\,\mathrm{kg\,m^{-2}yr^{-1}}$ level near site 2, but experience a strong gradient along the PIG main trunk.

Nearby ASIRAS observations at site 18 and 19 in particular suggest higher SMB values compared to the dated NP profiles. Site 19 is directly located at the centre of a pronounced accumulation trough of $\sim 2.5\,\mathrm{km}$ width, which adds to the uncertainty in the layer matching because of the spatial displacement between the iSTAR site and point of closest approach. Because the traced reflection layer significantly exceeds the depth range of the dated NP density profiles at site 18 and 19, we discarded both sites for the layer dating.

Additional local departures between our results and those from ME14 were identified for the northern slopes and southward interior of PIG. Because of difficulties in the layer tracing at the northern slopes, the authors of M14 had to augment their SMB estimates with results from a different layer, which they dated back to 2002 and corrected for a temporal bias to the 1985 layer based on overlapping segments. Thus, one possible explanation for the observed differences is that the true local temporal bias correction may be different from the regional scale bias correction, which they estimate from regression models. Other possible explanations are differences in the observational coverage and local accuracy from the different interpolation methods. With regard to the southward interior, the spatial coverage is superior by the ME14 results. Despite the maximum range limit between $100\,\mathrm{km}$ and $190\,\mathrm{km}$ for the ASIRAS based estimates, the missing observational constraints towards the interior may still yield an underestimation of the southward SMB gradient. However, we also cannot rule out that the smaller gradient in our observations is due to a local increase in SMB between the different observational periods among both studies. Additional observational constraints of the selected reflection layer may resolve the cause for the observed difference.

## 4.2 Elevation dependent model drift

The observational SMB estimates by M14 indicate an elevation dependent drift of simulated SMB from RACMO. The authors find that RACMO underestimates the SMB at the high-elevation interior, which would also impact our ASIRAS–RACMO based estimates of total mass input. Indeed, this finding is also reflected in our data (see supplement S1) and suggests that the ASIRAS–RACMO based total mass input estimates are biased by the underestimated SMB contribution from RACMO. According to (Agosta et al., 2019), the opposite may apply for the ASIRAS–MAR based estimates. The authors observe a tendency for MAR to overestimate accumulation on Ross-Marie Byrd Land and conclude that differences between MAR and RACMO2 are very likely related to differences in the advection inland. Similar to our elevation dependent comparsion between ASIRAS and RACMO SMB estimates, we find evidence of a drift in the MAR estimates with an opposite sign according to S1. We conclude that the best estimate for total mass input lies between ASIRAS–RACMO and ASIRAS–MAR estimates.

## 4.3 Impact on recent mass balance estimates

Despite the local differences in the SMB distribution, the difference between the $\Sigma_+$ estimates for the PIG catchment (including Wedge) between this study and ME14 is small, i.e. the ASIRAS–RACMO hybrid $\Sigma_+$ is $1.7\,\mathrm{Gt\,yr^{-1}}$ larger, which corresponds to 2% of the ME14 value. Similarly, the ASIRAS–MAR hybrid estimates are 5% larger compared to ME14, which is still

| Study | Period | Basin Definition | $\Sigma_+^-$ (Gt yr$^{-1}$) | Updated $\Sigma_+^-$ (Gt yr$^{-1}$) (RACMO;MAR) hybrid |
|---|---|---|---|---|
| M14 | 2005 - 2012 | Pine Island (M14) | $-41 \pm 7$ | $-(39;37) \pm 18$ |
| M14 | 2005 - 2010 | Wedge (M14) | $-0.6 \pm 2.5$ | $-(0.7;0.7) \pm 3.1$ |
| Gardner et al. (2018) | 2008 - 2015 | Zwally et al. (2012) | $-49 \pm 19$ | $-(40;38) \pm 28$ |
| Rignot et al. (2019) | 2005 - 2014 | Mouginot et al. (2017) | $-51 \pm 7$ | $-(46,44) \pm 20$ |

**Table 4.** Updated mass balance estimates $\Sigma_+^-$ for different studies based on hybrid $\Sigma_+$ estimates of this study using complementary (RACMO;MAR) SMB estimates. Indicated periods refer to the considered ice loss processes. Net gain from $\Sigma_+$ is assumed to be stationary during the indicated periods.

within the uncertainty range estimated by the authors of M14. This indicates that the local differences in the SMB estimates between both studies cancel out. If we take into account that the temporal averaging time used by ME14 is about a factor of 2.7 larger than that used in this study, we cannot find evidence of a potential secular trend in SMB at decadal scales similar to that of the ice discharge at PIG. This provides additional evidence to Medley et al. (2013) that the recent temporal evolution of the PIG mass balance is primarily driven by dynamic ice loss into the Amundsen Sea.

With regard to existing mass balance estimates for PIG, we have to take into account that basin outlines can differ significantly between studies as illustrated in Fig. 9 (d). To evaluate the impact of our hybrid SMB estimates on recent mass balance inventories, we extracted results from the literature in Tab. 4 and added updated mass balance estimates $\Sigma_+^-$ by replacing the $\Sigma_+$ estimates from the literature with the $\Sigma_+$ estimates of this study. We assume that the SMB remains stationary for the mass balance calculation with regard to the shown periods. In addition, we linearly interpolated the estimated ice discharge measurements in ME14 for the missing periods before 2007. Furthermore, we assume that the unspecified basin definitions in ME14 are in close agreement with the basin definitions based on Fretwell et al. (2013).

The small difference between the $\Sigma_+$ estimates of this study and ME14 directly translates into the $\Sigma_+^-$ mass balance estimates. The largest impact of our results is on the $\Sigma_+^-$ estimate by Gardner et al. (2018). After replacing their $\Sigma_+$ estimate from RACMO2.3 simulations with our ASIRAS–MAR hybrid $\Sigma_+$ estimate, the mass balance increases by $\sim 11$ Gt yr$^{-1}$.

## 4.4 SMB uncertainty

While the agreement in $\Sigma_+$ estimates between this study and ME14 supports the hypothesis that the regional SMB of PIG is stationary at decadal scales, our uncertainty estimates are much larger. The temporal error according to Fig. 4, which is $\sim 5\%$ larger than Medley et al. (2013) and ME14, cannot fully explain the difference between both uncertainty estimates. We also do not expect any major differences with regard to the spatial uncertainty of the density profiles. According to the error-grid statistics of the ASIRAS–RACMO based estimates in Tab. 5, we identify the backtransformed interpolation standard deviation $S_0$ from the OLK scheme as the dominating error source of our results, while the combined error in ME14 is slightly above our measurement error. The dominating $S_0$ uncertainty is also evident in Fig. 9 (a,b,c), where the spatial features of the combined

| % | Pine Island | | | | Wedge | | | |
|---|---|---|---|---|---|---|---|---|
| | $\mu_{err}$ | $\sigma_{err}$ | min | max | $\mu_{err}$ | $\sigma_{err}$ | min | max |
| Measurement Error | 9.9 (7.2) | 5.2 (1.8) | 3.7 | 40.5 | 7.2 | 0.9 | 5.8 | 9.5 |
| $S_0$ | 32.2 (25.5) | 32.7 (34.9) | 0.5 | 481.8 | 12.5 | 11.3 | 2.1 | 57.3 |
| Combined Error | 34.4 (27.4) | 32.4 (34.2) | 6 | 481.9 | 15.2 | 10.2 | 7.3 | 57.8 |
| Medley et al. (2014) | 10.4 | 6.1 | 2.6 | 30.0 | 6.3 | 1.8 | 2.9 | 10.8 |

**Table 5.** Mean $\mu_{err}$, standard deviation $\sigma_{err}$, minimum, and maximum gridded SMB errors with respect to basin wide hybrid ASIRAS–RACMO SMB estimates. Mean and $\sigma$ values in brackets are limited to the 190 km practical range threshold and weighted according to the ASIRAS partitioning in Fig. 8 (b).

error-grid are predominately determined by the $S_0$ grid. We find that the low accumulation zone at the northern slopes of PIG, which is next to the main trunk between iSTAR site 1 to 6, shows combined $S_0$ patches that considerably exceed 100%. In contrast, combined error estimates in ME14 do not exceed 20% at the same location.

Initial tests on our OLK setting revealed that the choice of the negative kriging weights correction method has a noticeable impact on the uncertainty estimates, a finding, which according to our knowledge, has not been reported before. However, our applied method by Deutsch (1996) already yields the minimum uncertainty estimates for our results, whereas the additional methods cited in Yamamoto (2000) yields an additional uncertainty increase between 20% (Froidevaux, 1993) and 50% (Journel and Rao, 1996).

Additional tests, where we used the kriging standard deviation based on non-transformed OK estimates, did not improve our interpolation uncertainty. Therefore the different choice of the interpolation uncertainty measure is not the source of the larger uncertainty range of this study. We hypothesize that despite the homoscedastic (i.e. data value independent) nature of the krige standard deviation, the reduction of data variance after subtracting the regression surface according to ME14 is most likely the cause of their significantly lower uncertainty estimates.

In addition to the larger uncertainty range of this study, we note that the choice between cell-by-cell summation and RSS of grid errors has a quite substantial impact on the $\Sigma_+$ uncertainty estimates. If we make the optimistic assumption that gridded errors are independent and choose the calculation of RSS instead, $\Sigma_+$ uncertainty estimates would reduce to $\pm 0.5 \, \mathrm{Gt \, yr^{-1}}$ (i.e. $\sim 97\%$ less) for the combined Pine Island and Wedge basin.

## 4.5 Systematic retrieval impacts

In addition to the uncertainty assessment in Sec. 4.4, we evaluated the impact of artificial cluster removal, the choice of permittivity model, and the non-transformed OK scheme.

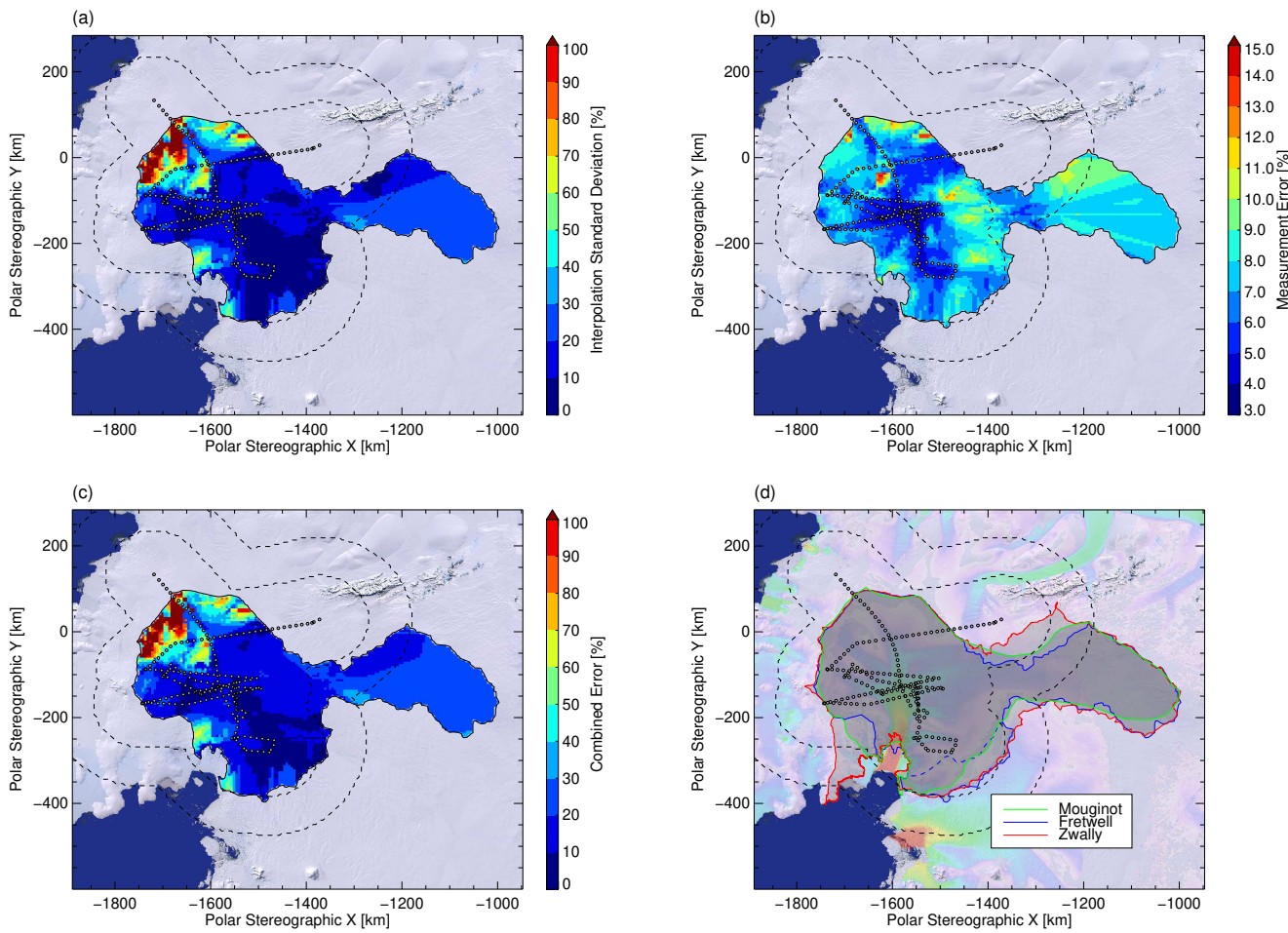

**Figure 9.** (a,b) Interpolation standard deviation $S_0$ and Measurement error of Fig. 8 (a), respectively. (c) Root sum square of (a,b). (d) Varying PIG basin definitions according to Zwally et al. (2012), Fretwell et al. (2013), and Mouginot et al. (2017). Surface flow speeds adopted from Fig. 1. Background imagery taken from U.S., Geological Survey (2007) for all panels.

### Artifical cluster removal

Inspection of the artificial cluster highlighted in Fig. 8 revealed that it is centred around the location with the lowest observed SMB and is essentially generated by the local nonbias terms of the OLK procedure. Owing to its steep contrast with the surroundings, it appears to be plausible to replace this cluster by averaged values of its nearest neighbours. However, due to the limited extend of this cluster, its additional contribution to the $\Sigma_+$ estimates would be less than 0.8%. Similarly, the impact on the PP-plot is negligible. Increasing the translational constant $C$ helps removing this cluster, but at the cost of statistical

agreement between observations and estimates.

**Looyenga based results**

Defining $\epsilon'$ by Eq. (2) instead of Eq. (1) yields a minor reduction of $\Sigma_+$ for the PIG catchment of 0.6%, which we expect from Sinisalo et al. (2013). However, despite the minor impact of the alternative definition for $\epsilon'$, we noticed an additional small impact on the layer dating, which shifted our estimated layer formation from November to September 2004. Thus, we had to adjust the time range in the RACMO SMB extraction for the calculation of hybrid SMB estimates. While the choice of the $\epsilon'$ model only has a minor impact on our total mass input estimates, it is worth noting that the effect on our annual layer dating is detectable.

**Non-transformed kriging results**

If we choose the OK procedure instead, $\Sigma_+$ increases by 4% for the Pine Island and 12% for the Wedge area, which would further increase the offset between this study and ME14. However, inspection of the SMB distribution (not shown) indicates that estimates tend to overshoot near the coastline of the Amundsen Sea, which becomes particularly evident for the Wedge area. Hence, the OK procedure appears to be more sensitive to the limited observational constraints near the Wedge area. In addition, $S_0$ based uncertainty estimates increase by 27% and 88% for the Pine Island and Wedge area, which highlights the improved performance of the OLK procedure.

## 5  Conclusions

Our analysis provides updated mean annual SMB estimates for the PIG basin and 2005–2014 averaging period based on a comprehensive airborne radar and ground truthing survey and complementary model simulations. Based on these estimates, we calculated a total mass input of $79.9\pm19.2\,\mathrm{Gt\,yr^{-1}}$ and $82.1\pm19.2\,\mathrm{Gt\,yr^{-1}}$ for the PIG basin area when using complementary RACMO and MAR SMB estimates, respectively. In comparison with earlier estimates from airborne radar observations, which consider the 1985–2009 averaging period, our results show a greater total mass input between 2% and 5%. This increase is still within the uncertainty range of both studies. Hence, no distinct trend is visible between both averaging periods and thus not compensating the accelerated ice discharge. We conclude that our results provide further evidence that the recent total mass input can be considered stationary at decadal scales. This implies that the increased dynamic ice loss over past decades remains the driving force in the recent mass balance evolution of PIG. However, departures between both observations at the northern slopes and southward interior of PIG, which cancel out for the estimates on total mass input, may indicate temporal changes in the local SMB distribution. Furthermore, our radar based observations can resolve a discrepancy between strainrate and SMB measurement at iSTAR site 2, which highlights the benefit of such complementary SMB measurements for future missions. Despite the minor changes in total mass input between both studies, the more than twofold uncertainty range of our results remains striking. Neither the applied model for the wave propagation speed of radar soundings nor the uncertainty related to the regional density profile can explain the larger uncertainty of this study. The same also applies for the reduced temporal averaging time. A comprehensive evaluation of our uncertainty estimation revealed that assumptions on the geostatistical in-

terpolation error as well as grid-error dependences can have a substantial impact on the uncertainty estimation. In terms of the error partitioning, our interpolation error is the dominating source of combined grid-errors. Moreover, varying basin definitions have an impact on our total mass input estimate by up to 19%. This highlights the importance of a thorough documentation

of uncertainty estimates and basin definitions to improve future intercomparisons between different SMB and mass balance inventories.

*Data availability.* https://doi.pangaea.de/10.1594/PANGAEA.XXXXXX

## Appendix A: List of Abbreviations and Notations

| | |
|---|---|
| ASIRAS | Airborne SAR / Interferometric Radar Altimeter System |
| GPR | Ground Penetrating Radar |
| M14 | Medley et al. (2014) |
| Combined Error | root-sum-of-squares of measurement and interpolation standard deviation |
| Measurement Error | root-sum-of-squares of spatial, temporal, and digitization error components |
| NP | Neutron Probe |
| OK | ordinary krige procedure |
| OLK | ordinary logarithmic kriging procedure |
| RACMO | RACMO2.3p2 regional climate model |
| PP-plot | Percentile–Percentile plot |
| RSS | root-sum-of-squares |
| SMB | Surface Mass Balance, $\mathrm{kg\,m^{-2}yr^{-1}}$ |
| T1 | iSTAR traverse 2013/14 |
| T2 | iSTAR traverse 2014/15 |
| TWT | Two-Way-Traveltime of radar soundings |
| PIG | Pine Island Glacier |
| w.e. | water equivalent |
| WAIS | West Antarctic Ice Sheet |
| | |
| $a$ | layer age, years |
| $\dot{b}$ | annual mean SMB, $\mathrm{kg\,m^{-2}yr^{-1}}$ |
| $\Delta D$ | closest distance between ASIRAS track and iSTAR site |
| $\epsilon'$ | real part of the dielectric permittivity |
| $N$ | number of reflection layer points considered for annual dating |
| $\rho$ | density, $\mathrm{kg\,m^{-3}}$ |
| $\mathrm{R_{max}}$ | maximum range threshold to ASIRAS measurements, $\mathrm{km}$ |
| $\sigma_x$ | standard deviation of $N$ layer dating estimates |
| $\Sigma_+$ | total mass input, $\mathrm{Gt\,yr^{-1}}$ |
| $\Sigma_+^-$ | total mass balance, $\mathrm{Gt\,yr^{-1}}$ |
| $S_0$ | interpolation standard deviation |

*Author contributions.* S.Kowalewski conceived of the presented idea, designed the computational framework, adapted and tested the geo-statistical krige methods, accomplished the reflection layer tracing in large parts, reprocessed the Neutron Probe density profiles for the data

calibration, wrote the manuscript with input from all authors, V. Helm performed the SAR level_1b ASIRAS data processing, provided the digital elevation model, and established access to the RACMO2.3p2 data, E. Morris delivered the Neutron Probe density profiles, and O. Eisen contributed to layer analysis and interpretation. All authors discussed the results and contributed to revising the manuscript.

*Competing interests.* Olaf Eisen is co-editor in chief of The Cryosphere

*Acknowledgements.* The German Ministry of Economics and Technology (grant 50EE1331 to S. Kowalewski) and European Space Agency (ESA) have supported this work as part of the CryoSat validation program. The authors gratefully acknowledge the excellent logistical support provided by British Antarctic Survey's (BAS) Rothera Research Station and members of the iSTAR Traverse and Alfred Wegener Institute during the field campaign, which has been funded by the UK Natural Environment Research Council, NERC grant NE/J005681/1.
The authors express their gratitude towards A. Shepherd, PI of iSTAR-D project, for his general support and data contribution to this study. The authors thank R. Mulvaney (BAS, UK) and H. Konrad (CPOM, University of Leeds, UK and DWD, Germany), for provision of datasets and discussions in an early stage. The authors gratefully acknowledge the provision of RACMO2.3p2 model output by S. Ligtenberg and M. van Wessem (IMAU, Utrecht University, NL). The authors would like to thank Emerson E&P Software, Emerson Automation Solutions, for providing licenses in the scope of the Emerson Academic Program. We sincerely appreciate the valuable comments and suggestions by the
referees and editor.

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
