# Peer review of "The regional scale surface mass balance of Pine Island Glacier, West Antarctica over the period 2005–2014, derived from airborne radar soundings and neutron probe measurements"

_The Cryosphere, 2020_

## Referee Comment (RC1) · Brooke Medley (Referee) · 18 Jul 2020

Kowalewski et al., The regional scale surface mass balance of Pine Island Glacier, West Antarctica over the period 2005–2014, derived from airborne radar soundings and neutron probe measurements

The authors present a very thorough and well conducted analysis of airborne radar data collected over the Pine Island Glacier catchment to recover estimates of surface mass balance. By tracking single horizon (circa 2005), dated through unique neutron

probe depth-age and depth-density information, they successfully mapped SMB over a large portion of the Pine Island catchment area. The radar technique presented is robust, and the authors apply an improved kriging technique to spatially interpolate the tracks to a more complete grid. The results are largely in agreement with a prior work, suggesting little change in snow accumulation over the region and also highlighting the robustness of the approach.

The paper details a significant amount of work that is praiseworthy. A few clarifications are required to improve transparency in a few locations (see Major Comments). The paper is generally well-written and outlined but could use substantial editing for improved flow and clarity. It is great to read such interesting studies; I have made a few major and minor comments below for the authors to consider to help improve the presented work.

Major Comments

- On line 49 there is mention of "adjust"ing layers at crossovers. How was this accomplished? Will this impact the results? In the description later on line 132-134, there should be discussion regarding the magnitude of the adjustments. Some discussion of what this means for the total uncertainty would be welcome. Are only the layer traces impacted or are the layer ages adjusted to the adjusted layer depth?

- Regarding the analysis of the NP density profiles: please clarify in the Fig. 2 caption an in the paragraph beginning Line 88 that only n = 22 profiles are used in the calculation of the depth-density profile? If the entire 43 individual profiles are used, it would act to minimize the standard deviations of the profile. Are the "43 NP profiles" in line 89 all from traverse T2?

- The authors comment that the picked layer depths (Lines 142-143) do not necessarily associate with a peak in the density profile yet argue that density-driven dielectric contrasts form the radar reflection horizons. Do the authors have any insight into why that might be? Some further discussion of this would strengthen the paper.

- Uncertainty Analysis: The authors clearly put in a significant amount of time designing the uncertainty analysis, which is quite commendable and appreciated. Some additional details would help clarify the exact plan since the authors produce two estimates (Fig 4 a and c), and it become unclear which is being used in what way. It is not entirely clear why there is a "picking" uncertainty in the age uncertainty as well as an evaluation of the standard deviation of the ages in Table 2. Wouldn't any uncertainty in picking layers manifest in the numbers in Table 2? There is also uncertainty in the NP dating; perhaps this is what the +/- 1 year uncertainty is meant for? What does the small-scale variability in snow accumulation look like? Can comparisons be made from over 2 km away? Perhaps a good alternative would be a simple weighted mean of the age by distance (as well as standard deviation). Please also clarify that in Eqn 5, the "spatial" component refers to the density profile. Finally, is it justifiable to assume that the spatial errors in density are uncorrelated with depth? While firn depth-density is noisy with depth, there is no reason to expect that under different accumulation/temperature regimes, there will be large biases between measurements (that will accumulate with depth, rather than cancel out).

Minor Comments

- Some description of the range of "N" in Section 2.4 would help clarify the robustness of the methodology. Maybe the authors could add the N to Table 2. Also, be consistent with significant digits. Why do some have 2 decimal places and others only one? Based on the remainder of the text, it looks like the standard is just 1 decimal place. Also, please explain the "comment"s in the Table 2 caption.

- Is the uncertainty in the depth of the tracked layer in determination of the age uncertainty accounted for? From Line 148, it sounds like only a standard deviation of points is used. Additional uncertainty is imposed from the translation of the uncertainty in depth to an uncertainty in age based on the slope of the depth-age profile.

- For Figure 4, please indicate that the darker shades of grey refer to a higher density

of layers. Also, the caption should have more information. It's difficult to attribute what calculations are made in Figure 4 c,d. Perhaps clarify that the "Spatial" error is in reference to the variability in density.

- Several studies have shown that RACMO is biased (often too low accumulation in the interior), so is it sufficient to take the RACMO values at face value when adding them to the regions that you cannot resolve. Perhaps a correction could be made to the RACMO data first using overlap between the radar estimates and the model over the region of overlap.

- The uncertainties provided are found to be larger than in ME14, which is expected as the authors note. From the text, the temporal error from this work is 1.4 y / 10.1 y = 13.9%, which indicates that the smallest error for any radar-derived SMB estimate is effectively 14%. In ME14, the temporal uncertainty is 1 y / 25 yr = 4%, suggesting that the temporal uncertainties are much lower in ME14. That reduction is likely due to the robust dating techniques used on the firn cores used in that study. Based on this alone, it is not clear how the shorter time window and larger dating uncertainties in this analysis do not at least account for a substantial amount of the increased basin-wide uncertainty values found in this work. Locally, it does appear that uncertainties from the kriging technique are much larger in this work than in ME14; however, they appear smaller than some of the uncertainties (say, in the southernmost reaches of the catchment) than in ME14. The total ME14 basin-wide uncertainty is 9% (6.8/78.3), whereas this work is 24% (19.2/79.9). Adding an additional 10% temporal uncertainty to the ME14 estimates puts values at 19%. Therefore, the likely impact of the kriging technique is on the order of 5%. This is a very simplistic take but should be robust from an order of magnitude perspective.

Minor Edits

Line 4: consider replacing "allowing" since "allow" is used in the previous sentence
Line 5: add a comma after "scheme" Line 6: replace "the same main" with "similar"

Line 17: remove "the" in front of "upwelling" Line 18: consider replacing "stimulating" with "initiating" Line 28: replace "in particularly" with "in particular" Line 29: clarify what is meant regarding the sentence starting with "Basin wide mass..." It is unclear what it means in this context and might require a citation. Line 30: replace "in the following" with "hereinafter" Lines 32-33: remove "the" before "logistical" Line 33: change to "dielectric properties of snow and firn" Line 37: replace "In the following," with "Hereinafter," Line 44: replace "trace" with "track" Line 48: replace "self-intersections" with "crossovers" Line 50: change "measurement" to "measurements"

Note, there are several minor fixes needed beyond section 1, which will require further refinement by the authors.

Figure 3: add a righthand y-axis with depth equivalent.

---

## Referee Comment (RC2) · Anonymous Referee #2 · 27 Aug 2020

This paper presents recent surface mass balance (SMB) estimates derived from airborne radar observations, and ground based neutron probe measurements of snow density along the iSTAR traverse (2013,2014) at Pine Island Glacier (PIG). This paper interestingly focuses on data uncertainties resulting from methodology and assumptions made on the interpolation error, demonstration and compares estimates with those given in previous publication from Medley et al., 2014, and with RACMO2.3p2 simulations. This paper is well written, presents an interesting new dataset for model validation and deserves to be published.

[Figure]

However, before publication, I have a few questions and suggestions. I hope these could improve the final paper.

Major comments:

1. My main comment concerns the dating of reflection layers. Here, the dating relies only on big assumptions made on stratigraphy characteristics. This technique is based on differences in winter and summer snow due to changes in atmospheric conditions and radiative fluxes. I suppose that dating of reflection layers and NP data is accurate but there is no comparison with stake networks, or with a clear "absolute" dating based on anthropogenic radionuclides or volcanic horizons at several cores. Since layers sometimes mismatch at several intersections, or are excluded (around iSTAR sites 2 and 19 for instance), the final dating may be not fully robust. I understand that the authors define a layer age uncertainty of +-1.4 year and assess the associated uncertainty in the surface accumulation, but is it possible that the layer dating uncertainty exceeds 1 year? In particular, melt or rain is expected to occur mainly in summer, but is it possible that significant surface melt (or rain) occurrences occurred at the beginning and end of summer but were separated by an "extreme" solid precipitation event (Turner et al., 2019), leading to a 2 maxima in density and in other snow characteristics used for the layer counting? Is there any snow erosion, which could remove the surface layer at locations in the area? Are there any stake farms or well constrained ice cores (with an absolute age of one or various layers) in the area, on which the authors could validate their estimates in snow accumulation?

2. Neutron probe is a really interesting way to retrieve snow density and the authors clearly took profit from this technique in the past and in the present paper. However, data rely on a few calibration steps. I had a look to previous papers from Konrad et al., 2019 and Morris et al., 2017, and I did not really understand how density data were validated before being used in the present paper. My concern is because in Figure 2 we observe that snow density is 550 kg m-3 at 7m below the ground level, whereas it is 600 kg m-3 at the same depth in Morris et al., 2017. Snow density in firn cores
from Konrad et al., 2019 are hard compare here because their Figure 3 is developed until 50 m. Could the author describe whether they calibrated the NP snow density data with snow pit data in the present paper or not? If not, is there a difference in the density/depth relationships between (Konrad et al., 2019), (Morris et al., 2017) and present paper. If snow density from NP measurements is biased, how will this impact the final SMB values?

3. Concerning the kriging method, is it worth using northing, easting, and elevation as explanatory variables? Would it be more relevant to use the distance from the coastline as explanatory variable? Or even the distance from Amundsen sea coast?

4. Comparison of ASIRAS data with RACMO2.3p2 simulations are really interesting but differences are not fully justified/explained in the text. Since differences are model dependant, it would be interesting to see potential differences with another model used in Antarctica (e.g., COSMO-CLM, HIRHAM5, MAR3.10, MetUM , see Mottram et al., 2020). Since Agosta et al. (2019) proposed potential justifications of the differences existing between RACMO2.3p2 and at least the MAR model, I believe that a comparison with the MAR model would make sense here. Indeed, in Agosta et al., 2019, large differences between RACMO and MAR are observed in regions where the RACMO2.3p2 model presents the largest differences with the ASIRAS data. A quick comparison could be relevant to discriminate whether the precipitation formation, advection of hydrometeors, and sublimation of precipitating hydrometeors (Agosta et al., 2019) are important or not in the PIG area. Data from Agosta et al. 2019 are available here: https://zenodo.org/record/2548848#.X0St8TXgphE If the authors are interested in higher resolution simulations, data from a more recent paper from Donat-Magnin et al., 2020, focusing in the Amundsen region are also available at: https://doi.org/10.5281/zenodo.2815907.

5. The paper largely describes differences with Medley et al. (2014) paper, but the authors never include any figure presenting the differences. I propose to include a map presenting ME14 route, SMB results and differences with the ASIRAS data.

6. The paper is sometimes quite hard to follow for a non-expert of this area. Different datasets are used here, and the difference between ISTAR/ASIRAS and ME14 is not always clear. The authors use a many acronyms. I suggest that the authors include a table where they clearly describe the difference between ASIRAS/iSTAR data used here, and more particularly the difference with the ME14. For instance, Table 1 presents different radars (ASIRAS, pulseEKKO PRO GPR , CReSIS Accu-R) part of the information is given in the caption, but perhaps the authors could also precise in the table if field campaigns were deployed on the ground or by plane, over what distance? which were the reflection layers used for SMB estimates? which density measurements were considered (NP? Firn cores?)? how was performed the dating of the reflection layers?

7. Figures could display the route followed by ME14, and where GPR from Konrad et al., 2019 was carried out.

Minor comments: Abstract

"Thus there is no evidence of a secular trend in mass input to the PIG basin." => please be more accurate because secular may be misinterpreted here. I suggest to replace secular (here and elsewhere) by decadal.

Line 28: "in particularly"

Lines 38-41: firn cores are only used to retrieve the depth of dated snow layer? Are they used to calibrate the neutron probe density data?

Figure 1. ASIRAS-iSTAR survey : please also include the location of ground GPR observations from Konrad et al., 2019?

Line 68, the authors write: " Due to the reported consistency between the GPR and airborne SMB measurements in Konrad et al. (2019), we limit the comparison of our results to the basin wide estimates by ME14". I feel that a figure (perhaps in the supplementary material) showing the different radargrams could help the reader. A quick

data comparison, before interpolation, could also be done to see how snow density and radar data uncertainty impact the final SMB value.

Line 85 "Morris et al. (2017) applied an automatic annual layer identification routine to their snow density profiles and used the annual $H_2O_2$ peak depths as an additional guidance for the annual layer dating." => is it possible to observe the removal of on year of snow in the PIG area (due to erosion) or multiple summer maxima?

Line 88: Here I understand that the authors did not consider the density obtained from the firn cores to compute the final SMB. What is the difference in the final SMB if the authors use the density from firn cores to calibrate the snow density profiles?

Line 89: 43 profiles => I suppose this means that profiles were done twice at the 22 sites?

Line 92: is there any relationship between snow density profile and Accumulation/Temperature as suggested by Herron & Langway 1980 equation?

Line 120: how does hoar produce thin ice layers? Is it possible to have short rain our melt events in spring or late summer? Could wind erosion or sublimation create wind crusts in this region?

Line 144: "We assign an estimated date to the mean value of N points" => do you mean "the mean value of depth"?

Line 198: "may by due"

Line 275: please discuss this sentence according to Agosta et al. (2019) results. Indeed, according to this publication, sublimation of precipitating hydrometeors are missed at low elevation in RACMO2.3p2. This could justify that RACMO 2.3p2 presents a positive bias at low elevation. Conversely, they suggest that MAR snowfall rates generally exceed those simulated by RACMO2.3p2, by more than 30% on the lee side of the West AIS (Marie Byrd Land toward Ross ice shelf), and close to crests at the ice sheet margins. Here, a comparison with MAR could be interesting.

Lines 288: The authors refer to the ASIRAS or the Hydrid SMB estimates, but line 296 they refer to the ASIRAS vs. the ME14 one, whereas they refer to the Hybrid estimate at line 298. The difference between these estimates is not clear. Why do the authors use ASIRAS / hydrid estimates in different parts of the text?

Line 325: "but also measured density and strainrate profiles suggest a mean annual SMB of 200 kgm−2yr−1 based on theoretical grounds, which both are in a better agreement with the collocated ASIRAS based results." => Is it possible to explain briefly the way this value of 200kg m-2 is computed?

Line 328- 331: please give elevation of sites 2, 18, 19. It would be interesting to discuss the differences between the RACMO2.3p2 and MAR models here (is there any potential explanation related to precipitation formation /advection /sublimation of precipitating particles to the ground described in both models).

Line 352: "secular" => decadal?

Line 395: "Inspection of the artificial cluster highlighted in Fig. 5 revealed" => replace by fig. 8?

Table 2: please include site coordinates and elevation.

Table 3: Are the basins from Mouginot et al. (2017) similar to those given by Rignot et al. (2019)? If not, perhaps it would make sense to use the most recent basins.

Table 5: do values refer to results written as OLK(NN) in Figure 7?

Figure 5: please include ISTAR site numbers on a) and b) to help the reader in retrieving the locations cited in the text.

Figure 7: Please define NN

Figure 8: I suggest to include the elevation contours. Is it correct to write Eastings when the x-axis is related to the north-south direction, and Northings for the y-axis when it is on the east/west direction? I don't understand why the kriging procedure

induce strange vertical (or horizontal) lines of similar values in Figure 8a (this point is particularly visible near the glacier outlet)? Is it because northing, easting, and elevation are used as explanatory variables? If yes, is it not more relevant to use the distance from the coastline as explanatory variable? Or even the distance from Amundsen sea coast?

References:

Agosta, C., Amory, C., Kittel, C., Orsi, A., Favier, V., Gallée, H., van den Broeke, M. R., Lenaerts, J. T. M., van Wessem, J. M., van de Berg, W. J., and Fettweis, X.: Estimation of the Antarctic surface mass balance using the regional climate model MAR (1979–2015) and identification of dominant processes, The Cryosphere, 13, 281–296, https://doi.org/10.5194/tc-13-281-2019, 2019.

Donat-Magnin, M., Jourdain, N. C., Gallée, H., Amory, C., Kittel, C., Fettweis, X., Wille, J. D., Favier, V., Drira, A., and Agosta, C.: Interannual variability of summer surface mass balance and surface melting in the Amundsen sector, West Antarctica, The Cryosphere, 14, 229–249, https://doi.org/10.5194/tc-14-229-2020, 2020.

Herron, M. M., and C. C. Langway, 1980: Firn Densification: An Empirical Model. J. Glaciol., 25, 373–385, doi:10.3198/1980JoG25-93-373-385.

Konrad, H., Hogg, A. E., Mulvaney, R., Arthern, R., Tuckwell, R. J., Medley, B., and Shepherd, A.: Observations of surface mass balance on Pine Island Glacier, West Antarctica, and the effect of strain history in fast-flowing sections, Journal of Glaciology, pp. 1–10, https://doi.org/10.1017/jog.2019.36, 2019.

Medley, B., Joughin, I., Smith, B. E., Das, S. B., Steig, E. J., Conway, H., Gogineni, S., Lewis, C., Criscitiello, A. S., McConnell, J. R., van den Broeke, M. R., Lenaerts, J. T. M., Bromwich, D. H., Nicolas, J. P., and Leuschen, C.: Constraining the recent mass balance of Pine Island and Thwaites glaciers, West Antarctica, with airborne observations of snow accumulation, The Cryosphere, 8, 1375–1392, https://doi.org/10.5194/tc-8-

1375-2014, 2014.

Morris, E. M., Mulvaney, R., Arthern, R. J., Davies, D., Gurney, R. J., Lambert, P., De Rydt, J., Smith, A. M., Tuckwell, R. J., and Winstrup, M.: Snow Densification and Recent Accumulation Along the iSTAR Traverse, Pine Island Glacier, Antarctica, Journal of Geophysical Research: Earth Surface, 122, 2284–2301, https://doi.org/10.1002/2017JF004357, 2017.

Mottram, R., Hansen, N., Kittel, C., van Wessem, M., Agosta, C., Amory, C., Boberg, F., van de Berg, W. J., Fettweis, X., Gossart, A., van Lipzig, N. P. M., van Meijgaard, E., Orr, A., Phillips, T., Webster, S., Simonsen, S. B., and Souverijns, N.: What is the Surface Mass Balance of Antarctica? An Intercomparison of Regional Climate Model Estimates, The Cryosphere Discuss., https://doi.org/10.5194/tc-2019-333, in review, 2020.

Turner, J., Phillips, T., Thamban, M.,Rahaman, W., Marshall, G. J.,Wille, J. D., et al. (2019). The dominantrole of extreme precipitation events inAntarctic snowfall variability.Geophysical Research Letters,46,3502–3511. https://doi.org/10.1029/2018GL081517

---

## Author Comment (AC1) · 14 Oct 2020

Author responses to Referee #1

**Major Comment 1 (MC1)**    "On line 49 there is mention of "adjust"ing layers at crossovers."

We agree that the term "adjusting" is not properly describing the manual layer "correction", which we did during the layer tracing.

"How was this accomplished?"

While we initially used the annual markers in the density profiles from the Neutron Probe (NP) measurements to trace the same isochronic reflection layer throughout the entire flight survey, we recognised crossover points where the traced layer did not match between the intersecting profiles. This was typically the case for profiles which are subject to complex stratigraphy or poor signal returns. As stated on Line 128, we performed a visual inspection of the vertical reflection layer sequence to guide the layer bridging wherever it is required. Intermediate reflection features or low signal-to-noise may challenge the visual inspection, which also depends on the experience in the manual layer tracing. We therefore considered layer mismatches at crossover sections as indicators for systematic errors that occurred during the manual bridging at difficult radargram sections. Typically, retracing the adjacent layer of one profile could fix the mismatch at one crossover point, but may yield a new mismatch at another crossover point. Hence, we were iteratively retracing challenging profile sections until we achieved a plausible intersection of the same layer at all crossover points.
We can clarify this point according to the explanation above in the revised text and replace the term "adjusting" with "correcting".

"Will this impact the results?"

If we selected the reflection layer based on the annual markers at iSTAR sites only, we would shift the ASIRAS results towards a better agreement with the iSTAR results at their closest approach. However, this would certainly result in inconsistent trace jumps at crossover points. Hence, the applied correction makes the  ASIRAS estimates independent from single iSTAR sites, which we also point out on Line 134.

"In the description later on line 132-134, there should be discussion regarding the magnitude of the adjustments. Some discussion of what this means for the total uncertainty would be welcome"

It is difficult to reconstruct the impact of these corrections to the results because they were continuously applied during the manual layer tracing with the aim of producing a consistent trace for the selected reflection layer as the final product.

"Are only the layer traces impacted or are the layer ages adjusted to the adjusted layer depth?"

The layer age of the final trace is estimated based on the depth age scales at all iSTAR sites (Table 2). It is correct that the allocated layer age changes with the new depth of a retraced (i.e. corrected) layer. We think that this point becomes clear, once we explained the manual correction as discussed above in the revised text.

**MC2**    "Regarding the analysis of the NP density profiles: please clarify in the Fig. 2 caption an in the paragraph beginning line 88 that only n = 22 profiles are used in the calculation of the depth-density profile? If the entire 43 individual profiles are used, it would act to minimize the standard deviations of the profile. Are the "43 NP profiles" in line 89 all from traverse T2?"

We agree that the stated number of 43 NP profile scan lead to some confusion, as also questioned in mC8 of Referee #2. According to our reply, we count the adjacent profile sections at each site (apart from site 2) separately on Line 89. These profiles were concatenated to a single profile per site (no concatenation necessary at site 2). Indeed, Figure 2 already shows the concatenated 21 profiles in addition to the single profile of site 2, which we shall clarify in the revised text.

**MC3**   The authors comment that the picked layer depths (Lines 142-143) do not necessarily associate with a peak in the density profile yet argue that density-driven dielectric contrasts form the radar reflection horizons. Do the authors have any insight into why that might be? Some further discussion of this would strengthen the paper.

We suppose that local noise in the stratigraphy is one obvious explanation for the observed differences between reflection layer and density peak depths. Keeping in mind that we are comparing point measurements with spatially integrated radar profiles (4.5 m along track bin, according to Table 1), which are separated by several 10s or 100s of metres (with one even being 2 km), the resulting dating uncertainty from the ensemble of site matches (Table 2) is rather good and my not be expected from areas with low accumulation (Line 314).

Another explanation is a difference in the timing of the formation of the reflection layer and density peak, i.e. the dielectric contrast generated by the formation of thin ice layers (Arcone et al., 2004 and 2005) may not coincide with density peaks.

This question is closely related to MC1 of the Referee #2 and will be considered as discussed above in the methods section.

**MC4**   "Uncertainty Analysis: The authors clearly put in a significant amount of time designing the uncertainty analysis, which is quite commendable and appreciated. Some additional details would help clarify the exact plan since the authors produce two estimates (Fig 4 a and c), and it become unclear which is being used in what way."

Figures 4 a and c show two different results according to Line 182-185, where we say that we consider the error estimates based on Fig. 4 (c and d) for the measurement uncertainty estimation. We can add this to the caption.

"It is not entirely clear why there is a "picking" uncertainty in the age uncertainty as well as an evaluation of the standard deviation of the ages in Table 2. Wouldn't any uncertainty in picking layers manifest in the numbers in Table 2? There is also uncertainty in the NP dating; perhaps this is what the +/- 1 year uncertainty is meant for?"

We agree that the term "picking uncertainty" is confusing with regard to the additional evaluation of layer intercepts at iSTAR sites as pointed out above. Indeed, we shall use "layer dating uncertainty" instead and clarify the reasoning for this uncertainty in the text as follows.

While other studies such as Medley et al. (2013) consider an uncertainty of +/-1 month for the formation of the annual reflection layer, we extend this uncertainty range to one year to account for systematic errors in the picking process which may arise from ambiguous reflection features or bridging of difficult sections along the flight track (see also our response to MC1, Referee #2). Hence, we are using the same uncertainty range for the parameter $\overline{\delta t}$ = +/- 1 year (Line 158) as in Konrad et al. (2019) but for a different reasoning (i.e. +/- 1 year uncertainty to account for the 1 year offset between GPR and firn-core data).

"What does the small-scale variability in snow accumulation look like?"

We assume that this question refers to the lateral variability in the stratigraphy (Line 159) at the range of displacement between the iSTAR location and ASIRAS point of closest approach. Typically the stratigraphy is not flat at the iSTAR locations and may contain pronounced gradients which are visible in the radar imagery. We agree that the question is justified as to whether a layer dating based on measurements separated by 2 km in one case is reliable. We included this extreme case in the analysis as the stratigraphy was rather flat according to the radar imagery. We suppose that the reliability of such comparisons strongly depends on the degree of folding in the stratigraphy, which is quite variable along the flight track. Even though we are limited to the flight direction (Line 149), we consider the standard deviation $\sigma_x$ (Line 148) as a benchmark for the uncertainty which results from the lateral folding in the stratigraphy between the closest ASIRAS measurements and iSTAR locations. As proposed in our reply to mC5 of Referee #2, we plan to make the radar data available at PANGAEA to allow for a further inspection of small scale variability.

"Perhaps a good alternative would be a simple weighted mean of the age by distance (as well as standard deviation)."

The proposed alternative sounds reasonable, but we would not expect a noticeable impact on the results due to the already small $\sigma_x$ values according to Table 2. As we are limited in our assessment of layer folding towards the flight direction, we think that it is reasonable to make a more conservative assumption by giving all age estimates along the flight track the same weight.

"Please also clarify that in Eqn 5, the "spatial" component refers to the density profile."

Yes, we shall clarify this in the text.

Finally, is it justifiable to assume that the spatial errors in density are uncorrelated with depth? While firn depth-density is noisy with depth, there is no reason to expect that under different accumulation/temperature regimes, there will be large biases between measurements (that will accumulate with depth, rather than cancel out).

We assume for Eq.(5) that errors are uncorrelated (Line 171). This also includes spatial errors with depth as questioned above. However, we replace the root-sum-squares (RSS) of spatial error terms with their absolute values to estimate the measurement uncertainty for our results. This implies that different errors cannot cancel each other out (Line 182), i.e. we are considering the most pessimistic scenario by this assumption. Any error correlation with depth will yield to an uncertainty equal or below this uncertainty threshold value. In this connection, we are over- rather than underestimating the measurement error by neglecting the covariance terms in Eq.(5) that account for any spatial error correlation with depth. Therefore, we think that this question should already be answered on Lines (181-185).

**Minor Comment (mC1)**     "Some description of the range of "N" in Section 2.4 would help clarify the robustness of the methodology. Maybe the authors could add the N to Table 2."

Yes, we can add the range of N to Table 2 to give a better sense of the considered sample sizes.

"Also, be consistent with significant digits. Why do some have 2 decimal places and others only one? Based on the remainder of the text, it looks like the standard is just 1 decimal place."

The number of significant digits was selected based on the precision of lateral uncertainty and therefore we stick to the least significant digit, which varies.

"Also, please explain the "comment"s in the Table 2 caption."

In short, "extrapolated" means that the reflection layer exceeds the depth of the deepest annual marker in the NP density profile at this site, hence, the layer age has been estimated based on extrapolation of the depth age scale. We discarded these measurements because we cannot assess the robustness of these extrapolated values. "Noise" means that no stratigraphy was visible in the profile section, e.g. due to exceeding rolling angles of the plane. For the erroneous NP dating, we refer to Line 320-326.

We can include the explanations above in the revised Table 2 caption.

**mC2**, "Is the uncertainty in the depth of the tracked layer in determination of the age uncertainty accounted for? From Line 148, it sounds like only a standard deviation of points is used. Additional uncertainty is imposed from the translation of the uncertainty in depth to an uncertainty in age based on the slope of the depth-age profile."

We confirm that we only consider the standard deviation of $N$ age estimates near each site to estimate the layer uncertainty from the spatial displacement of the ASIRAS and NP measurements. We agree that any uncertainty in the TWT to depth conversion translates to an additional uncertainty in layer age. However, in contrast to the considered uncertainty for the SMB estimates (Eq. 5), we are not using the fitted regional density profile from all iSTAR sites, but we use the local iSTAR density profile for the layer dating at each site (Line 140). Thus, we do not have to consider the spatial uncertainty of the composed regional density profile and expect that the error in the TWT to depth conversion is dominated by the digitization error, which we expect to have a minor impact on the resulting uncertainty.

**mC3** "For Figure 4, please indicate that the darker shades of grey refer to a higher density of layers. Also, the caption should have more information. It's difficult to attribute what calculations are made in Figure 4 c,d. Perhaps clarify that the "Spatial" error is in reference to the variability in density."

We can include the suggested clarifications for the caption of Fig. 4 in the revised manuscript.

**mC4** "Several studies have shown that RACMO is biased (often too low accumulation in the interior), so is it sufficient to take the RACMO values at face value when adding them to the regions that you cannot resolve. Perhaps a correction could be made to the RACMO data first using overlap between the radar estimates and the model over the region of overlap."

We have to test the suggested correction first and may consider its impact on total mass input in the revision. Depending on the impact of MAR results on the discrepancy above, we may also consider its impact on total mass input (see response to MC4, Referee #2).

**mC5** The uncertainties provided are found to be larger than in ME14, which is expected as the authors note. From the text, the temporal error from this work is 1.4 y / 10.1 y

= 13.9%, which indicates that the smallest error for any radar-derived SMB estimate is effectively 14%. In ME14, the temporal uncertainty is 1 y / 25 yr = 4%, suggesting that the temporal uncertainties are much lower in ME14. That reduction is likely due to the robust dating techniques used on the firn cores used in that study. Based on this alone, it is not clear how the shorter time window and larger dating uncertainties in this analysis do not at least account for a substantial amount of the increased basinwide uncertainty values found in this work.

We agree that the smaller layer age uncertainty and the larger temporal averaging interval reduce the temporal uncertainty in M14 noticeably. Even if we assumed the same layer age uncertainty of 1.4 y for M14, the resulting temporal uncertainty of 1.4/25 = 5,6% would still remain noticeably smaller compared to this study. It is difficult though to give an expectation about its impact on the measurement error grid based on the percentages mentioned above. The displayed error grids in M14 are limited to the combined error, hence, we lack information about the spatial distribution of the partitioning between the interpolation and measurement errors in M14. Furthermore, both studies use different Krige methodologies which use different expressions for the interpolation error (Line 234-240) and may differ in their response to added noise to the SMB input values.

Locally, it does appear that uncertainties from the kriging technique are much larger in this work than in ME14; however, they appear smaller than some of the uncertainties (say, in the southernmost reaches of the catchment) than in ME14. The total ME14 basin-wide uncertainty is 9% (6.8/78.3), whereas this work is 24% (19.2/79.9). Adding an additional 10% temporal uncertainty to the ME14 estimates puts values at 19%. Therefore, the likely impact of the kriging technique is on the order of 5%. This is a very simplistic take but should be robust from an order of magnitude perspective.

Indeed, a local cluster of high interpolation errors North of the flight track between iSTAR 1 to 6 (Line 375) is significantly higher compared to the combined errors in M14. Care must be taken though when comparing the combined error grid maps due to the considered larger bin range of this study [0%,100%) and M14 [0%,30%]. For instance, the second and third colour shades (i.e. [10,20]% and [20,30%] ) of this study already reach the maximum colour bin of [25,30]% of M14 to the southernmost reaches of the catchment.

**Minor Edits**

Line 4: consider replacing "allowing" since "allow" is used in the previous sentence
Line 5: add a comma after "scheme" Line 6: replace "the same main" with "similar"
Line 17: remove "the" in front of "upwelling" Line 18: consider replacing "stimulating" with "initiating" Line 28: replace "in particularly" with "in particular" Line 29: clarify what is meant regarding the sentence starting with "Basin wide mass..." It is unclear what it means in this context and might require a citation. Line 30: replace "in the following" with "hereinafter" Lines 32-33: remove "the" before "logistical" Line 33: change to "dielectric properties of snow and firn" Line 37: replace "In the following," with "Hereinafter," Line 44: replace "trace" with "track" Line 48: replace "self-intersections" with "crossovers" Line 50: change "measurement" to "measurements"
Note, there are several minor fixes needed beyond section 1, which will require further refinement by the authors.
Figure 3: add a righthand y-axis with depth equivalent.

We appreciate the suggested minor edits and consider them for the revision.

Arcone, S., Spikes, V., Hamilton, G., & Mayewski, P. (2004). Stratigraphic continuity in 400MHz short-pulse radar profiles of firn in West Antarctica. Annals of Glaciology, 39, 195-200. doi:10.3189/172756404781813925

Arcone, S., Spikes, V., & Hamilton, G. (2005). Phase structure of radar stratigraphic horizons within Antarctic firn. Annals of Glaciology, 41, 10-16. doi:10.3189/172756405781813267

Konrad, H., Hogg, A., Mulvaney, R., Arthern, R., Tuckwell, R., Medley, B., & Shepherd, A. (2019). Observations of surface mass balance on Pine Island Glacier, West Antarctica, and the effect of strain history in fast-flowing sections. Journal of Glaciology, 65(252), 595-604. doi:10.1017/jog.2019.36

Medley, B. et al. (2013), Airborne-radar and ice-core observations of annual snow accumulation over Thwaites Glacier, West Antarctica confirm the spatiotemporal variability of global and regional atmospheric models, Geophys. Res. Lett., 40, 3649– 3654, doi:10.1002/grl.50706.

---

## Author Comment (AC2) · 14 Oct 2020

Author responses to Referee #2

**Major Comment 1 (MC1)** "My main comment concerns the dating of reflection layers. Here, the dating relies only on big assumptions made on stratigraphy characteristics. This technique is based on differences in winter and summer snow due to changes in atmospheric conditions and radiative fluxes. I suppose that dating of reflection layers and NP data is accurate but there is no comparison with stake networks, or with a clear "absolute" dating based on anthropogenic radionuclides or volcanic horizons at several cores. Since layers sometimes mismatch at several intersections, or are excluded (around iSTAR sites 2 and 19 for instance), the final dating may be not fully robust. I understand that the authors define a layer age uncertainty of +-1.4 year and assess the associated uncertainty in the surface accumulation, but is it possible that the layer dating uncertainty exceeds 1 year? In particular, melt or rain is expected to occur mainly in summer, but is it possible that significant surface melt (or rain) occurrences occurred at the beginning and end of summer but were separated by an "extreme" solid precipitation event (Turner et al., 2019), leading to a 2 maxima in density and in other snow characteristics used for the layer counting? Is there any snow erosion, which could remove the surface layer at locations in the area? Are there any stake farms or well constrained ice cores (with an absolute age of one or various layers) in the area, on which the authors could validate their estimates in snow accumulation?"

We agree that the concerns in MC1 about the interpretation of measured density peaks and internal reflection layers as annual markers are justified.

We propose to address the concerns in MC1 in the methods sections for the NP and ASIRAS data as follows:

Formation of melt layers can be produced at PIG in response to melt events. We shall include a reference to Scott et al. (2010), who observed an exceptional ice layer at lower PIG at 20 m depth. However, no other ice layers (> 1mm thick) were observed at their discussed core site. We shall also include an additional reference to Nicolas et al. (2017) to address the potential impact of large warm air intrusion events to the West Antarctic ice sheet on surface melt. In this connection, we expect that large scale surface melt episodes at PIG are connected to the specific coupling of atmospheric modes and therefore do not follow a frequent seasonal pattern. We cannot rule out, though, that seasonal surface melt events are more common near the coastline and therefore may result in the formation of ambiguous annual peaks in the NP and ASIRAS profiles, which we shall include in the text of the method sections (see also our response to MC4 of Referee #1).

Morris et al. (2017,Sect. 4) provide a detailed discussion on the applied methods for interpretation of annual density markers from the iSTAR NP measurements, which we shall refer to in the NP methods section of this study. We shall also refer to Morris, 2008 with regard to the applied calibration equation of NP measurement, which is based on theoretical considerations and validated using laboratory experiments and gravimetric data from ice cores. This calibration was performed independently from the iSTAR cores.

Surface snow erosion or missing formation of annual markers cannot be ruled out for the NP data according to Morris et al. (2017) and for the ASIRAS data. For the latter, we are pointing out on Line 128 that bridging of data becomes necessary where annual layers cannot be traced, e.g. due to wind erosion. This may also result in initial mismatches between the traced layer at some intersection, which we attempted to correct by retracing adjacent layers (see our reply to MC1 of Referee #1). Therefore, it is important to outline that we aimed at producing a consistent layer trace

which matches at all intersections. We shall make this point more clear by using the term "correcting" instead of adjusting in the text, as proposed in our reply to Referee #1.

In terms of the robustness of our layer dating approach, we benefit from the intercomparison of different annual markers (i.e. reflectivity, density, and H2O2 at core locations) at iSTAR sites, which obey different physical processes in their formation. The photochemical H2O2 tracer (Line 81) follows a clear annual cycle in the PIG cores, which we can add to the text. Wind erosion may remove peaks in the H2O2 profile. This is less of a concern where surface accumulation is high. Locations exposed to pronounced wind erosion are visible in the observed stratigraphy from the ASIRAS soundings (Line 131), but annual reflection layers remain distinguishable at iSTAR locations, which supports the assumption of undisturbed formation of annual H2O2 markers here. According to internal communication with the iSTAR drilling team, no absolute volcanic reference horizon was accessible for the processed iSTAR cores. Stake farms do not exist for the considered region, but single stake measurement were performed between T1 and T2.

With regard to the assumptions made on stratigraphic characteristics of the observed internal reflection layers, we propose to refer to additional earlier studies in the introduction section (Line 30): The work of Arcone et al. (2004 and 2005) provides theoretical and observational evidence of the formation of continuous isochronal reflection layers throughout West Antarctica.

The question "is it possible that the layer dating uncertainty exceeds 1 year?" should have already been addressed in Sect. 2.4. It is true that we cannot completely rule out the possibility of systematic errors in the layer dating which may result from ambiguous annual signals or unintended layer changes in the manual tracing, but according to our response to MC4 of Referee #1 we shall clarify that an associated layer dating uncertainty of +/-1 years has already been taken into account for the estimated $\Delta t = 1.4$ years uncertainty of the layer age. It is highly unlikely that such systematic error persists for all data samples (i.e. NP, core measurements, and ASIRAS) at all sites which we use for the estimation of $\Delta t$ and so we consider the error estimation above to be reasonable in terms of the robustness of the layer dating approach of this study.

**MC2** "Neutron probe is a really interesting way to retrieve snow density and the authors clearly took profit from this technique in the past and in the present paper. However, data rely on a few calibration steps. I had a look to previous papers from Konrad et al., 2019 and Morris et al., 2017, and I did not really understand how density data were validated before being used in the present paper. My concern is because in Figure 2 we observe that snow density is 550 kg m-3 at 7m below the ground level, whereas it is 600 kg m-3 at the same depth in Morris et al., 2017. Snow density in firn cores from Konrad et al., 2019 are hard compare here because their Figure 3 is developed until 50 m. Could the author describe whether they calibrated the NP snow density data with snow pit data in the present paper or not? If not, is there a difference in the density/depth relationships between (Konrad et al., 2019), (Morris et al., 2017) and present paper. If snow density from NP measurements is biased, how will this impact the final SMB values?"

We are afraid that different units for depth, i.e. water equivalent depth (Fig. 2, Morris et al. 2017) and geometric depth (Fig. 2, this study) led to some confusion. Conversion of geometric to water equivalent depth yields an approximate density of 600 kg m^-3 at site 21 and 7 m level for this study. Even though, some of the referenced earlier studies use water equivalent scales, we prefer to keep SI units in this study to avoid potential pitfalls from the conversion to a specific reference scale as these are unambiguous. Nonetheless, the difference is well spotted by the referee and highlights the required caution to be taken when comparing different scales.

**MC3**   "Concerning the kriging method, is it worth using northing, easting, and elevation as explanatory variables? Would it be more relevant to use the distance from the coastline as explanatory variable? Or even the distance from Amundsen sea coast?"

Our initial motivation was to follow the methodology by M14 as close as possible for our data comparison. It is an interesting idea, if changes to the parametrisation of the regression model improved the statistical characteristics of residual SMB to the regression surface in our case. However, limiting the explanatory variables to the coastal distance would also limit the degrees of freedom in the regression model. We can only speculate at this point, but orographic effects in response to atmospheric circulation patterns may not be adequately captured by such a parametrisation.

Another potential explanatory variable may be temperature (Arthern et al., 2006), which affects saturation vapor pressure, which again correlates with accumulation rates. In turn, any correlation between elevation as explanatory variable in M14 and temperature makes both variables interchangeable for the regression model.

However, due to the persisting skewness of the residual SMB values based on our regression (Line 197), we completely rejected this approach for our analysis and used a logarithmic data transformation of the SMB samples instead (Line 202). It seems that this point is not clearly pointed out in our description and will be added accordingly.

**MC4**   "Comparison of ASIRAS data with RACMO2.3p2 simulations are really interesting but differences are not fully justified/explained in the text. Since differences are model dependant, it would be interesting to see potential differences with another model used in Antarctica (e.g., COSMO-CLM, HIRHAM5, MAR3.10, MetUM , see Mottram et al., 2020). Since Agosta et al. (2019) proposed potential justifications of the differences existing between RACMO2.3p2 and at least the MAR model, I believe that a comparison with the MAR model would make sense here. Indeed, in Agosta et al., 2019, large differences between RACMO and MAR are observed in regions where the RACMO2.3p2 model presents the largest differences with the ASIRAS data. A quick comparison could be relevant to discriminate whether the precipitation formation, advection of hydrometeors, and sublimation of precipitating hydrometeors (Agosta et al., 2019) are important or not in the PIG area. Data from Agosta et al. 2019 are available here: https://zenodo.org/record/2548848#.X0St8TXgphE If the authors are interested in higher resolution simulations, data from a more recent paper from Donat-Magnin et al., 2020, focusing in the Amundsen region are also available at: https://doi.org/10.5281/zenodo.2815907."

Even though this study uses model data to augment missing observations, its main emphasis is on the observations and applied methodologies to provide a means for validation to the model community. While MAR may perform better in some regards, RACMO may do so in others. From the standpoint of an observational study this is difficult to judge and already exceeds the scope of this study. We are also concerned that the inclusion of a model-to-model comparison will impact its conciseness (see also MC6). However, if the MAR model helps reduce the elevation depended offset to our observations, we may include this point in a revised version of Figure 9 and refer to the findings above.

**MC5**   "The paper largely describes differences with Medley et al. (2014) paper, but the authors never include any figure presenting the differences. I propose to include a map presenting ME14 route, SMB results and differences with the ASIRAS data."

We are afraid that the inclusion of further maps showing the M14 SMB results and differences with the ASIRAS data would make the paper less concise and may not contribute to its readability as concerned in MC6. Any side-by-side comparison between both maps is still possible based on the published results in M14 and the main focus of our intercomparison is on the estimated total mass input between both studies, which is considered in the provided tables of this study.

However, we think that it is useful to include the flight tracks of M14 in Figure 1 to aid any side-by-side comparison of the regional SMB distribution between both studies. If access to the flight coordinates is granted by the authors of M14, we include the flight tracks in the revised manuscript version.

**MC6**  "The paper is sometimes quite hard to follow for a non-expert of this area. Different datasets are used here, and the difference between ISTAR/ASIRAS and ME14 is not always clear. The authors use a many acronyms. I suggest that the authors include a table where they clearly describe the difference between ASIRAS/iSTAR data used here, and more particularly the difference with the ME14. For instance, Table 1 presents different radars (ASIRAS, pulseEKKO PRO GPR , CReSIS Accu-R) part of the information is given in the caption, but perhaps the authors could also precise in the table if field campaigns were deployed on the ground or by plane, over what distance? which were the reflection layers used for SMB estimates? which density measurements were considered (NP? Firn cores?)? how was performed the dating of the reflection layers?"

We appreciate the suggested improvements on the readability of the manuscript and further clarification of considered data products and we consider these suggestions for the revision.

**MC7**  "Figures could display the route followed by ME14, and where GPR from Konrad et al., 2019 was carried out."

As suggested in our response to MC5 we include the flight track followed by M14 in Figure 1 of the revised version, if data access is possible.

The ASIRAS flight track follows the iSTAR traverse, including the GPR measurements (Fig. 4 in Konrad et al., 2019), in most parts. We therefore prefer to limit the displayed routes to datasets, which we compare in this study to maintain readability of included maps.

**Minor Comment 1 (mC1)**  "Thus there is no evidence of a secular trend in mass input to the PIG basin." => please be more accurate because secular may be misinterpreted here. I suggest to replace secular (here and elsewhere) by decadal."

We agree that any statement about secular trends depends on the considered timescale and needs clarification in the text. It is our intention to express that we could not find evidence of a pronounced shift between both studies which may be of non-periodic nature in response to the steady acceleration of dynamic ice loss of PIG since the 1970s. We think that replacing the term secular with "decadal" would be more limiting in this regard, therefore we prefer to clarify our intention for using this term in the text (Line 352).

**mC2**  "in particularly"

correct spelling, "in particular"

**mC3**   "Lines 38-41: firn cores are only used to retrieve the depth of dated snow layer? Are they used to calibrate the neutron probe density data?"

As mentioned in our reply to MC1, firn cores were not considered for the calibration of neutron probe measurements. However, firn cores were used to guide interpretation of the records at two sites.

**mC4**   "Figure 1. ASIRAS-iSTAR survey : please also include the location of ground GPR observations from Konrad et al., 2019?"

See our response to MC7

**mC5**   "Line 68, the authors write: " Due to the reported consistency between the GPR and airborne SMB measurements in Konrad et al. (2019), we limit the comparison of our results to the basin wide estimates by ME14". I feel that a figure (perhaps in the supplementary material) showing the different radargrams could help the reader. A quick data comparison, before interpolation, could also be done to see how snow density and radar data uncertainty impact the final SMB value."

We agree that a comparison of radargrams from different platforms can be useful for an ongoing discussion on the error assumptions made by different studies. However, we are cautious about the expectation that this may be done by means of a quick data comparison as suggested in mC5. The radar studies of M14 and Konrad et al. (2019) use a different reflection layer for their analysis compared to this study, which already impacts the uncertainty of their SMB estimates. Local noise in the stratigraphy varies along the flight track and impacts related uncertainties in addition. Our error discussion builds upon the developed methodologies in M14 and  Konrad et al. (2019), and already considers impacts from the uncertainty in the assumed regional density profile and dating precision for the selected reflection horizons. Figure 10 (a) already gives a benchmark for other studies to the estimated measurement uncertainty of our results. However, we think that it is of help for potential future studies to add the ASIRAS radargrams, interpolated, and non-interpolated reflection layer traces in units of annual SMB and TWT to the PANGAEA repository.

**mC6**   "Morris et al. (2017) applied an automatic annual layer identification routine to their snow density profiles and used the annual H2O2 peak depths as an additional guidance for the annual layer dating." => is it possible to observe the removal of on year of snow in the PIG area (due to erosion) or multiple summer maxima?"

See our response to MC1, where we address the possibility of wind removal of annual markers and multiple summer maxima in response to melt events.

**mC7**   "Here I understand that the authors did not consider the density obtained from the firn cores to compute the final SMB. What is the difference in the final SMB if the authors use the density from firn cores to calibrate the snow density profiles?"

We confirm that we only use density profiles from NP measurements for the estimation of the regional density depth relation (see our response to MC1). We consider that the NP calibration has already been thoroughly elaborated in Morris, 2008 (see our response to MC1). Indeed, iSTAR NP profiles were compared with the core profiles to check that there was no systematic difference between them. Hence, we expect that the impact of reducing the number of sites will be larger to the fit of the regional density profile than the difference between the collocated NP and core measurements.

**mC8**  "Line 89: 43 profiles => I suppose this means that profiles were done twice at the 22 sites?"

All NP density profiles are composed by two adjacent profiles per site, with the exception of site 2 (Line 78). We agree that counting the adjacent profiles separately may confuse the reader (i.e. 2x21 adjacent profiles plus the single profile at site 2) and we shall only count the resulting concatenated profiles in the revised text.

**mC9**  "Line 92: is there any relationship between snow density profile and Accumulation/Temperature as suggested by Herron & Langway 1980 equation?"

There is a relation between snow density profiles and accumulation and temperature, which is fully discussed in Morris et al. (2017). While the Herron & Langway 1980 (H&L) equation gives good predictions for stage 1 and stage 2 densification at the iSTAR sites, large departures to the H&L equation are observed for the transition zone between stage 1 and 2 and requires the consideration of a new model equation as described in Morris et al. (2017).

**mC10**  "Line 120: how does hoar produce thin ice layers? Is it possible to have short rain our melt events in spring or late summer? Could wind erosion or sublimation create wind crusts in this region?"

The first question "how does hoar produce thin ice layers" should already be covered by the provided reference to Arcone et al. (2004) on Line 120, i.e."[...] a hoar layer, which frequently occurs beneath thin ice layers because it is the source of vapor that creates the ice layer"

See our response to MC1 with regard to the possibility of rain or melt events at PIG.
With regard to the final question, "Could wind erosion or sublimation create wind crusts in this region?", wind crusts are more often observed in region with lower accumulation, e.g. on the polar plateau, than in a coastal area like PIG, where synoptic precipitation prevails.

**mC11**  "We assign an estimated date to the mean value of N points" => do you mean "the mean value of depth"?

Well spotted, the word "depth" is missing and will be added in the revised text.

**mC12**  "Line 198: "may by due""

Thanks, will replace "by" with "be"

**mC13**  "Line 275: please discuss this sentence according to Agosta et al. (2019) results. Indeed, according to this publication, sublimation of precipitating hydrometeors are missed at low elevation in RACMO2.3p2. This could justify that RACMO 2.3p2 presents a positive bias at low elevation. Conversely, they suggest that MAR snowfall rates generally exceed those simulated by RACMO2.3p2, by more than 30% on the lee side of the West AIS (Marie Byrd Land toward Ross ice shelf), and close to crests at the ice sheet margins. Here, a comparison with MAR could be interesting."

We agree that our discussion benefits from a reference to the findings of Agosta et al. (2019) and will be included in the revised text. To keep the scope of this study concise, we may prefer to limit our analysis to the RACMO based simulations as discussed in our response to MC4.

**mC14** Lines 288: The authors refer to the ASIRAS or the Hydrid SMB estimates, but line 296 they refer to the ASIRAS vs. the ME14 one, whereas they refer to the Hybrid estimate at line 298. The difference between these estimates is not clear. Why do the authors use ASIRAS / hydrid estimates in different parts of the text?

Line 296 should refer to hybrid estimates as well. We will add the word "hybrid" accordingly.

**mC15** "Line 325: "but also measured density and strainrate profiles suggest a mean annual SMB of 200 kgm−2yr−1 based on theoretical grounds, which both are in a better agreement with the collocated ASIRAS based results." => Is it possible to explain briefly the way this value of 200kg m-2 is computed?"

The value of 200 kg m^-2 was computed from the Herron and Langway Stage 1 equation based on the measured density and strainrate profile. We can address this in the revised text.

**mC16** "Line 328- 331: please give elevation of sites 2, 18, 19. It would be interesting to discuss the differences between the RACMO2.3p2 and MAR models here (is there any potential explanation related to precipitation formation /advection /sublimation of precipitating particles to the ground described in both models)."

We can add elevations for sites 2, 18, and 19. While the suggested model comparison sounds exciting, it appears to us better suited for a separate model intercomparison study, which may also include high resolution SMB measurements for comparison. (see also our replies to MC4 and mC13)

**mC17** "Line 352: "secular" => decadal?"

See our reply to mC1.

**mC18** "Inspection of the artificial cluster highlighted in Fig. 5 revealed" => replace by fig. 8?"

Well spotted, it should have been Figure 8

**mC19** "Table 2: please include site coordinates and elevation."

Due to the redundancy of site numbers in Table 2, the inclusion of site coordinates and elevation would yield to considerable white space. We suppose to refer to Table 1 in Morris et al. (2017) in the caption instead. All requested information is presented here.

**mC20** "Table 3: Are the basins from Mouginot et al. (2017) similar to those given by Rignot et al. (2019)? If not, perhaps it would make sense to use the most recent basins."
Yes, they are. See Rignot et al. (2019): "The […] drainage basis are available at National Snow and Ice Data Center (NSIDC), Boulder, CO as MEaSURES-2 products". The Mouginot et al. (2017) reference corresponds to the requested citation for using the data provided by NSIDC.

**mC21** "Table 5: do values refer to results written as OLK(NN) in Figure 7?" and
**mC23** "Figure 7: Please define NN"

No, OLK(NN) refers to the PP-plot by comparing *observations* against the nearest neighbour estimates to the *observations*. Ideally, probability density functions should agree between both of

them, which becomes evident by the 1:1 match in the PP-plot. However, if we only consider nearest neighbour *estimates* for the ASIRAS measurements, we would still miss a large fraction of the PIG catchment. This is why we have to expand our estimates to points beyond nearest neighbours of *observation*. By extending the allowed range of *estimates* to the *observations,* which they are based from, the PP-plot should still remain a straight line, which is the case for the applied ordinary lognormal kriging estimation. The indicated 100 km and 190 km ranges indicate the transition zone from estimates based on observations only and estimates based on RACMO.

Indeed, we missed to define NN as nearest neighbours in the text. This should also clarify the role of varying sample populations for Figure 7.

**mC22** "Figure 5: please include ISTAR site numbers on a) and b) to help the reader in retrieving the locations cited in the text."

We can do this for the revised version.

**mC24** "Figure 8: I suggest to include the elevation contours."

We initially tested elevation contours on top of Fig. 8 but we had the impression that it affected the readability of the images. We may, however, test larger increments between contour lines in a revised version of Figure 8.

"Is it correct to write Eastings when the x-axis is related to the north-south direction, and Northings for the y-axis when it is on the east/west direction?"

Eastings and northings are a common convention for geographic Cartesian coordinates and can be found for this region with the same orientation in the literature (e.g. Figure 2 in Grima et al., 2014). However, we agree that the northing/easting convention can be confusing when compared to actual directions. Another possibility, which we can do is to replace the axes annotations with "polar stereographic x [km]" and "polar stereographic y [km]" to avoid any confusion about directions.

"Is it because northing, easting, and elevation are used as explanatory variables? If yes, is it not more relevant to use the distance from the coastline as explanatory variable? Or even the distance from Amundsen sea coast?"

We spot a remaining misunderstanding in this question, i.e. "[…] is it not more relevant to use the distance from the coastline as explanatory variable?" In contrast to ME14, we do not use a regression model to produce residual SMBs for the kriging method (see our reply to MC3).

Indeed, straight lines are indicative of artifacts in the interpolation scheme, which limits the estimation to a fixed number of nearest neighbours as suggested by Yamamoto, 2007.
Again, these neighbours are split up according to the quadrant criterion, which can produce such artifacts depending on the number of nearest neighbours. We tested increasing the number of nearest neighbours, which reduces these artifacts but yield to a departure between *observations* and *estimations* in the PP-plot, which we use as a benchmark for the conversation of statistical properties during the logarithmic data transformation.

Arcone, S., Spikes, V., Hamilton, G., & Mayewski, P. (2004). Stratigraphic continuity in 400MHz short-pulse radar profiles of firn in West Antarctica. Annals of Glaciology, 39, 195-200. doi:10.3189/172756404781813925

Arcone, S., Spikes, V., & Hamilton, G. (2005). Phase structure of radar stratigraphic horizons within Antarctic firn. Annals of Glaciology, 41, 10-16. doi:10.3189/172756405781813267

Arthern, R. J., Winebrenner, D. P., and Vaughan, D. G. (2006), Antarctic snow accumulation mapped using polarization of 4.3-cm wavelength microwave emission, *J. Geophys. Res.,* 111, D06107, doi:10.1029/2004JD005667.

Grima, C., D. D. Blankenship, D. A. Young, and D. M. Schroeder (2014), Surface slope control on firn density at Thwaites Glacier, West Antarctica: Results from airborne radar sounding, Geophys. Res. Lett., 41, 6787–6794, doi:10.1002/2014GL061635.

Konrad, H., Hogg, A., Mulvaney, R., Arthern, R., Tuckwell, R., Medley, B., & Shepherd, A. (2019). Observations of surface mass balance on Pine Island Glacier, West Antarctica, and the effect of strain history in fast-flowing sections. Journal of Glaciology, 65(252), 595-604. doi:10.1017/jog.2019.36

Morris, E. M. (2008), A theoretical analysis of the neutron scattering method of measuring snow and ice density, J. Geophys. Res., 113, F03019, doi:10.1029/2007JF000962.

Morris, E. M., Mulvaney, R., Arthern, R. J., Davies, D., Gurney, R. J., Lambert, P.,… Winstrup, M. (2017). Snow densification and recent accumulation along the iSTAR traverse, Pine Island Glacier, Antarctica. Journal of Geophysical Research: Earth Surface, 122, 2284 – 2301. https://doi.org/10.1002/2017JF004357

Nicolas, J., Vogelmann, A., Scott, R. et al. January 2016 extensive summer melt in West Antarctica favoured by strong El Niño. Nat Commun 8, 15799 (2017). https://doi.org/10.1038/ncomms15799

Scott, J., Smith, A., Bingham, R., & Vaughan, D. (2010). Crevasses triggered on Pine Island Glacier, West Antarctica, by drilling through an exceptional melt layer. Annals of Glaciology, 51(55), 65-70. doi:10.3189/172756410791392763

Yamamoto, J.K. On unbiased backtransform of lognormal kriging estimates. Comput Geosci 11, 219–234 (2007). https://doi.org/10.1007/s10596-007-9046-x

---

## Author Response (AR1)

**Final author response to referees 1 and 2 for**
**"The regional scale surface mass balance of Pine Island Glacier, West Antarctica over the period 2005–2014, derived from airborne radar soundings and neutron probe measurements" by S.Kowalewski et al.**

*The Cryosphere Discuss.*
*https://doi.org/10.5194/tc-2020-102*

We thank Dr. Brooke Medley and the second anonymous referee for their thoughtful comments and suggestions to improve the manuscript. We respond to each of the referee comments below. Line numbers given in our response refer to the discussion version of the manuscript prior to the revision and our response being in plain text.

Stefan Kowalewski
* * *
Response to referee 1, Brooke Medley

*The authors present a very thorough and well conducted analysis of airborne radar data collected over the Pine Island Glacier catchment to recover estimates of surface mass balance. By tracking single horizon (circa 2005), dated through unique neutron probe depth–age and depth-density information, they successfully mapped SMB over a large portion of the Pine Island catchment area. The radar technique presented is robust, and the authors apply an improved kriging technique to spatially interpolate the tracks to a more complete grid. The results are largely in agreement with a prior work, suggesting little change in snow accumulation over the region and also highlighting the robustness of the approach.*

*The paper details a significant amount of work that is praiseworthy. A few clarifications are required to improve transparency in a few locations (see Major Comments). The paper is generally well-written and outlined but could use substantial editing for improved flow and clarity. It is great to read such interesting studies; I have made a few major and minor comments below for the authors to consider to help improve the presented work.*

*Major Comments*

**(MC1)** - *On line 49 there is mention of "adjust"ing layers at crossovers. How was this accomplished? Will this impact the results? In the description later on line 132–134, there should be discussion regarding the magnitude of the adjustments. Some discussion of what this means for the total uncertainty would be welcome. Are only the layer traces impacted or are the layer ages adjusted to the adjusted layer depth?*

We agree that "adjusting layers" can be misleading in this context. We changed line 49 from

"In addition, the ASIRAS flight track contains several self-intersections so that we can compare and adjust traced layers at those points." to

"In addition, the ASIRAS flight track contains several crossovers which we used to validate the same isochronal reflector from different directions."

*How was this accomplished?*

Previous version: "We compared the traced layer at 34 intersections and 8 nearby flight track segments. In case of layer mismatches, the layer tracing was iteratively adjusted to find a consistent layer for the entire flight track."

We clarified this point in the new version as follows: "We checked the traced layer for possible mismatches at 34 crossover points and 8 nearby flight track segments, which may have resulted from systematic errors in the initial manual layer tracing.

Such mismatches were particularly observed along challenging profile sections and fixed by retracing the reflection layers which yield the best match at the crossover points."

*Will this impact the results? In the description later on line 132-134, there should be discussion regarding the magnitude of the adjustments. Some discussion of what this means for the total uncertainty would be welcome. Are only the layer traces impacted or are the layer ages adjusted to the adjusted layer depth?*

We think that our previous terminology, i.e. "adjusting layers", led to some confusion at this point. As outlined in the revised version of Line 132, our corrections refer to potential errors in the initial layer tracing, which we fixed after the inspection of crossover points. There is no remaining "adjustment" to the final layer trace, which would impact our SMB results.

**(MC2)** - *Regarding the analysis of the NP density profiles: please clarify in the Fig. 2 caption an in the paragraph beginning Line 88 that only n = 22 profiles are used in the calculation of the depth-density profile? If the entire 43 individual profiles are used, it would act to minimize the standard deviations of the profile. Are the "43 NP profiles" in line 89 all from traverse T2?*

As discussed on line 89, we are merging the shallow NP profile with the nearby deeper profile to generate a single density depth profile at the considered iSTAR site. However, we agree that counting both profiles as single entities can be confusing. Therefore, we removed the counting number "43" from the text and consider the merged density depth profiles as single entities in the revised text.

**(MC3)** - *The authors comment that the picked layer depths (Lines 142-143) do not necessarily associate with a peak in the density profile yet argue that density–driven dielectric contrasts form the radar reflection horizons. Do the authors have any insight into why that might be? Some further discussion of this would strengthen the paper.*

We agree that this point must be clarified in the text, as it is important to differentiate between the annual density modulation in the NP profiles and formation of hoar and ice layers, which generate a pronounced dieletric contrast due to their density contrast to the surrounding firn matrix. First, we changed Line 116 from

"We assume that the annual peaks in density lead to reflection layers because of the associated dielectric constant (Eisen et al., 2003)." to

"We assume that internal reflection layers are generated by the dielectric contrast at embedded thin ice and hoar layers (Arcone et al., 2004, 2005) and that these layers are formed at regional scales around summer/autumn (Medley et al., 2013). These layers may coincide with the annual density modulation, which we observe with the NP measurements."

We removed the following Lines 118-122 and shifted the discussion of intra-annual layers to the following Measurement Error section. On line 136 we discuss our assumptions made on stratigraphic characteristics (see our response to the first major comment of Referee 2).

**(MC4)** - *Uncertainty Analysis: The authors clearly put in a significant amount of time designing the uncertainty analysis, which is quite commendable and appreciated. Some additional details would help clarify the exact plan since the authors produce two estimates (Fig 4 a and c), and it become unclear which is being used in what way.*

This should be clear from the revised caption of Figure 4 (see our response to mc3) and we added the following sentence to Line 187:

"We consider the combined measurement error based on Fig. 4 (c-d) for our SMB estimates."

*"It is not entirely clear why there is a "picking" uncertainty in the age uncertainty as well as an evaluation of the stan-*

*dard deviation of the ages in Table 2. Wouldn't any uncertainty in picking layers manifest in the numbers in Table 2? There is also uncertainty in the NP dating; perhaps this is what the +/- 1 year uncertainty is meant for?"*

We agree that the reasoning behind the "picking" uncertainty requires clarification in the text. According to our revised "Measurement error estimation" section, the +/- 1 year uncertainty is meant for the potential removal of an annual layer or addition of an intra-annual layer (see our response to MC1 of Referee 2).

*What does the small-scale variability in snow accumulation look like? Can comparisons be made from over 2 km away?*

We assume that this question refers to the lateral variability in the stratigraphy (Line 159) at the range of displacement between the iSTAR location and ASIRAS point of closest approach. Typically the stratigraphy is not flat at the iSTAR locations and may contain pronounced gradients which are visible in the radar imagery. We agree that the question is justified as to whether a layer dating based on measurements separated by 2 km in one case is reliable. We included this extreme case in the analysis as the stratigraphy was rather flat according to the radar imagery. We suppose that the reliability of such comparisons strongly depends on the degree of folding in the stratigraphy, which is quite variable along the flight track. Even though we are limited to the flight direction (Line 149), we consider the standard deviation $\sigma_x$ (Line 148) as a benchmark for the uncertainty which results from the lateral folding in the stratigraphy between the closest ASIRAS measurements and iSTAR locations. As proposed in our reply to mC5 of Referee 2, we plan to make the radar data available at PANGAEA to allow for a further inspection of small scale variability.

*Perhaps a good alternative would be a simple weighted mean of the age by distance (as well as standard deviation).*

The suggested alternative sounds reasonable, but we do not expect a noticeable impact on the results due to the already small $\sigma_x$ values according to Table 2. As we are limited in our assessment of layer folding towards the flight direction, we think that it is reasonable to make a more conservative assumption by giving all age estimates along the flight track the same weight.

*Please also clarify that in Eqn 5, the "spatial" component refers to the density profile.*

We may already refer to Line 187:

"It is evident from Eq. (4) that the spatial uncertainty of the density profile affects both the integration depth and incremental mass."

which explains that the spatial term in Eq. 5 relates to the variability in density. In addition, we relate the spatial error to the variability in density in the new caption of Fig. 4 (see our reponse to mc3).

*Finally, is it justifiable to assume that the spatial errors in density are uncorrelated with depth? While firn depth-density is noisy with depth, there is no reason to expect that under different accumulation/temperature regimes, there will be large biases between measurements (that will accumulate with depth, rather than cancel out).*

We assume for Eq.(5) that errors are uncorrelated (Line 171). This also includes spatial errors with depth as questioned above. However, we replace the root-sum-squares (RSS) of spatial error terms with their absolute values to estimate the measurement uncertainty for our results. This implies that different errors cannot cancel each other out (Line 182), i.e. we are considering the most conservative scenario by this assumption. Any error correlation with depth will yield to an uncertainty equal or below this uncertainty threshold value. In this connection, we are over- rather than underestimating the measurement error by neglecting the covariance terms in Eq.(5) that account for any spatial error correlation with depth. Therefore, we think that this question should already be answered on Lines (181-185).

*Minor Comments*

**(mc1)** - *Some description of the range of "N" in Section 2.4 would help clarify the robustness of the methodology. Maybe the authors could add the N to Table 2.*

We added N values to Table 2

*Also, be consistent with significant digits. Why do some have 2 decimal places and others only one? Based on the remainder of the text, it looks like the standard is just 1 decimal place.*

The number of significant digits was selected based on the precision of lateral uncertainty and therefore we stick to the least significant digit, which varies.

*Also, please explain the "comment"s in the Table 2 caption.*

Old caption, " Dating (Year) with associated uncertainty of closest ASIRAS reflection layer to ith iSTAR site. "Track number" refers to the ASIRAS flight track naming convention and "$\Delta D$" is the closest distance between the ASIRAS track and the iSTAR site. Years in brackets are discarded from the regional layer age estimation for the reasons given in the Comment column."

New caption, "Dated reflection layer year at nearby iSTAR sites. "Track number" refers to the ASIRAS flight track naming convention (year of measurement season [4 digits], measurement type [1 digit], profile segment number [3 digits]), "$\Delta D$" is the closest distance between the ASIRAS track and iSTAR site, and N is the number of picking samples considered for layer dating. Years in brackets are discarded from the regional layer age estimation as follows: Significant departure between traced layer and depth–age scale at site 2 (see main text), layer gaps due to high noise levels in the radargram, layer is significantly exceeding the dated NP profile depth (values in brackets indicate extrapolated depth–age values).

**(mc2)** - *Is the uncertainty in the depth of the tracked layer in determination of the age uncertainty accounted for? From Line 148, it sounds like only a standard deviation of points is used. Additional uncertainty is imposed from the translation of the uncertainty in depth to an uncertainty in age based on the slope of the depth–age profile.*

Indeed, we are considering the standard deviation from N layer age estimates around each site for the given $2\Delta D$ interval. However, we are also considering the local density profile for the TWT to depth conversion (Line 140) and as such, the scattering follows the slope of the associated depth-age scale.

We noticed that we may determine the layer age at each site by averaging its $N$ layer age estimates instead of calculating a mean depth for all points and relate this value to an age value. We tested and implemented the former method in the revised version but it didn't have an impact on the final layer age of a = 10.1+-1.4.

We revised the Lines 139-151. Old version: "At N points along the closest approach of the ASIRAS track to each iSTAR site we estimated the depth of the reflection layer from its TWT, using the fitted density profile from that site. We also tested the use of the measured density profile for the TWT conversion instead, but we find that the impact on our results is negligible similar to Morris et al. (2017). The estimated depths do not necessarily coincide with the depth of a peak in the density profile i.e with the start of a mass balance year. We assign an estimated date to the mean value of N points by interpolation using the depth–age scale from the iSTAR site. These points are centred around the point of closest approach to the nearby iSTAR GPS location. We chose the associated interval length to be twice the distance $\Delta D$ between the iSTAR GPS location and point of closest approach of the flight track. Finally, we added six months to the estimated dates according to the mass balance year definition. This yields the estimated Year values for the reflection layer at each iSTAR site in Tab. 2. Following Konrad et al. (2019) we express the dating uncertainty as the standard deviation $\sigma_x$ in the N measurements. In this sense, our error estimate is more conservative than the standard error of the mean. Furthermore, we assume that the uncertainty due to local variation in

the stratigraphy is isotropic, which does not generally need to be true. However, according to Tab. 2 in most cases the overall impact of this effect is one order of magnitude smaller than the variability of dated years among all iSTAR sites."

New version: "Based on annual density markers, we can relate the snow and firn depth at each iSTAR site to its associated age and determine the reflection layer age from its depth at each cross section. Here, we use an exponential fit of the local density–depth profile for the TWT to depth conversion. The lateral displacement $\Delta D$ between the point of closest approach of the flight track and iSTAR site adds to the reflection layer dating uncertainty. We therefore consider all $N$ points which lie within a $2\Delta D$ interval along the flight track and which is centred at the point of closest approach for the layer dating. Based on the local depth–age scale, we relate the estimated depths of $N$ points to their ages and assign the final layer date to their mean value. To account for the mass balance year definition above, we add six months to the mean layer date, which is listed for all iSTAR positions in Tab. 2. In addition, we estimate the dating uncertainty from the $N$ lateral estimates by their standard deviation $\sigma_x$. In this sense, our error estimate is more conservative than the standard error of the mean. Furthermore, we assume that the uncertainty due to local variation in the stratigraphy is isotropic, which does not generally need to be true. However, according to Tab. 2 the overall impact of this effect is one order of magnitude smaller than the variability of layer age values among all iSTAR sites in most cases."

**(mc3)** - *For Figure 4, please indicate that the darker shades of grey refer to a higher density of layers. Also, the caption should have more information. It's difficult to attribute what calculations are made in Figure 4 c,d. Perhaps clarify that the "Spatial" error is in reference to the variability in density.*

We changed the old caption, "Spatial, temporal, digitization, and combined relative errors (left panels) and error partitioning (right panels). Grey background shades indicate the depth distribution of the traced 2005 reflection layer. (a,b) Based on error propagation according to Eq.(5). (c,d) Excluded error cancellation for the spatial error terms (see text)."

as follows: "Spatial, temporal, digitization, and combined SMB measurement errors, which relate to the variability in density, dating uncertainty, and ASIRAS sampling accuracy respectively: Relative errors (left panels) and error partitioning (right panels). Grey background shades indicate the depth distribution of the traced 2005 reflection layer with higher number concentrations towards darker shaded. (a,b) Based on error propagation according to Eq.(5). (c,d) Excluded spatial error cancellation in Eq.(5) [see main text] and considered for the final error estimation of this study."

**(mc4)** - *Several studies have shown that RACMO is biased (often too low accumulation in the interior), so is it sufficient to take the RACMO values at face value when adding them to the regions that you cannot resolve. Perhaps a correction could be made to the RACMO data first using overlap between the radar estimates and the model over the region of overlap.*

Despite the significant correlation, which we found between the elevation and SMB bias, the correlation remains week and as such, one may question as to whether a correction based on a week correlation is justified. The inclusion of MAR results to the revised manuscript should at least provide an alternative estimate, which helps to quantify the impact of an elevation dependent bias to our estimates.

**(mc5)** - *The uncertainties provided are found to be larger than in ME14, which is expected as the authors note. From the text, the temporal error from this work is 1.4 y / 10.1 y = 13.9%, which indicates that the smallest error for any radar-derived SMB estimate is effectively 14%. In ME14, the temporal uncertainty is 1 y / 25 yr = 4%, suggesting that the temporal uncertainties are much lower in ME14. That reduction is likely due to the robust dating techniques used on the firn cores used in that study. Based on this alone, it is not clear how the shorter time window and larger dating uncertainties in this analysis do not at least account for a substantial amount of the increased basin-wide uncertainty values found in this work.*

We agree that the smaller layer age uncertainty and the larger temporal averaging interval reduce the temporal uncertainty in M14 noticeably. Even if we assumed the same layer age uncertainty of 1.4 y for M14, the resulting temporal uncertainty of 1.4/25 = 5,6% would still remain noticeably smaller compared to this study. It is difficult though to give an expectation about its

impact on the measurement error grid based on the percentages mentioned above. The displayed error grids in M14 are limited to the combined error, hence, we lack information about the spatial distribution of the partitioning between the interpolation and measurement errors in M14. Furthermore, both studies use different Krige methodologies which use different expressions for the interpolation error (Line 234-240) and may differ in their response to added noise to the SMB input values.

*Locally, it does appear that uncertainties from the kriging technique are much larger in this work than in ME14; however, they appear smaller than some of the uncertainties (say, in the southernmost reaches of the catchment) than in ME14. The total ME14 basin-wide uncertainty is 9% (6.8/78.3), whereas this work is 24% (19.2/79.9). Adding an additional 10to the ME14 estimates puts values at 19%. Therefore, the likely impact of the kriging technique is on the order of 5%. This is a very simplistic take but should be robust from an order of magnitude perspective.*

Indeed, a local cluster of high interpolation errors North of the flight track between iSTAR 1 to 6 (Line 375) is significantly higher compared to the combined errors in M14. Care must be taken though when comparing the combined error grid maps due to the considered larger bin range of this study [0%,100%) and M14 [0%,30%]. For instance, the second and third colour shades (i.e.[10%,20%] and [20%,30%] ) of this study already reach the maximum colour bin of [25%,30%] of M14 to the southernmost reaches of the catchment.

*Minor Edits*

*Line 4: consider replacing "allowing" since "allow" is used in the previous sentence* We replaced "Ground based neutron probe measurements of snow density at 22 locations allow us to derive SMB from the annual internal radar reflection layers. The 2005 layer was traced for a total distance of 2367 km allowing us to determine annual mean SMB for the period 2005–2014."

with "Ground based neutron probe measurements provide information of snow and firn density with depth at 22 locations and were used to date internal annual reflection layers. The 2005 layer was traced for a total distance of 2367 km to determine annual mean SMB for the period 2005–2014."

*Line 17: remove "the" in front of "upwelling" Line 18: consider replacing "stimulating" with "initiating" Line 28: replace "in particularly" with "in particular".* Line 30: replace "in the following" with "hereinafter" Lines 32-33: remove "the" before "logistical. Line 33: change to "dielectric properties of snow and firn. Line 37: replace "In the following," with "Hereinafter,". Line 44: replace "trace" with "track" Line 48: replace "self-intersections" with "crossovers" Line 50: change "measurement" to "measurements. Figure 3: add a righthand y-axis with depth equivalent.

We applied all suggested corrections.

*Line 29: clarify what is meant regarding the sentence starting with "Basin wide mass..." It is unclear what it means in this context and might require a citation.*

We replaced "mass balance estimates" with "total mass input"

*Note, there are several minor fixes needed beyond section 1, which will require further refinement by the authors.*

Several minor fixes, e.g. restructuring of phrases, correcting grammar errors, etc. were applied beyond section 1.

Response to anonymous referee 2

*This paper presents recent surface mass balance (SMB) estimates derived from airborne radar observations, and ground based neutron probe measurements of snow density along the iSTAR traverse (2013,2014) at Pine Island Glacier (PIG). This paper interestingly focuses on data uncertainties resulting from methodology and assump- tions made on the interpolation error, demonstration and compares estimates with those given in previous publication from Medley et al., 2014, and with RACMO2.3p2 simulations. This paper is well written, presents an interesting new dataset for model validation and deserves to be published.*

*However, before publication, I have a few questions and suggestions. I hope these could improve the final paper.*

*Major Comments*

**(MC1)** - *My main comment concerns the dating of reflection layers. Here, the dating relies only on big assumptions made on stratigraphy characteristics. This technique is based on differences in winter and summer snow due to changes in atmospheric conditions and radiative fluxes. I suppose that dating of reflection layers and NP data is accurate but there is no comparison with stake networks, or with a clear "absolute" dating based on anthropogenic radionuclides or volcanic horizons at several cores.*

There were no stake networks available for comparison. We added to Line 85:

"According to internal communication with the iSTAR drilling team, no absolute volcanic reference horizon was accessible for the processed iSTAR cores and therefore limits their annual markers to the H2O2 and density profiles"

*Since layers sometimes mismatch at several intersections, or are excluded (around iSTAR sites 2 and 19 for instance), the final dating may be not fully robust.*

This directly relates to the concerns of referee 1 in her first major comment. As outlined in our response, we clarified in the new version that initial layer mismatches were caused by systematic errors in the manual layer tracing, which were corrected during the tracing process.

As requested by referee 1 in her first minor comment, we added explanations for the excluded iSTAR sites in the text.

*I understand that the authors define a layer age uncertainty of +-1.4 year and assess the associated uncertainty in the surface accumulation, but is it possible that the layer dating uncertainty exceeds 1 year? In particular, melt or rain is expected to occur mainly in summer, but is it possible that significant surface melt (or rain) occurrences occurred at the beginning and end of summer but were separated by an "extreme" solid precipitation event (Turner et al., 2019), leading to a 2 maxima in density and in other snow characteristics used for the layer counting? Is there any snow erosion, which could remove the surface layer at locations in the area? Are there any stake farms or well constrained ice cores (with an absolute age of one or various layers) in the area, on which the authors could validate their estimates in snow accumulation?*

We added the following text on line 136 to discuss the mentioned concerns about systematic errors in the layer dating:

"So far, we assumed that internal reflection layers form on an annual basis during summer/autumn but the potential formation of intra-annual reflection layers may challenge this assumption. For instance, Nicolas et al. (2017) found evidence of surface melt episodes over large parts of WAIS in response to warm air intrusion events. Scott et al. (2010) observed a strong reflection layer, which coincides with an exceptional melt layer at 22 m depth at one PIG ice core location. While such findings appear to be of sporadic nature at the basin scale and may be related to the coupling between atmospheric modes (Nicolas et

al., 2017), we find evidence of intra-annual reflection layers at coastward iSTAR sites where the snow accumulation is high. Extreme solid precipitation events may also impact the density modulation with depth (Turner et al., 2019), which is considered for the depth–age scale based on the NP measurements. Snow erosion may remove annual markers where accumulation rates are low. In addition to annual layer counting errors, the timing between the reflection layer formation and snow densification may be offset during summer/autumn. All these factors challenge the tracing and dating of the 2005 reflection layer but combining the stratigraphic information from the ASIRAS and iSTAR observations helps reducing the risk of systematic errors from erroneous layer counting. To account for the remaining risk in terms of isochronal accuracy, we assign an annual layer tracing uncertainty of $\overline{\delta t} = \pm 1$ years."

**(MC2)** - *Neutron probe is a really interesting way to retrieve snow density and the authors clearly took profit from this technique in the past and in the present paper. However, data rely on a few calibration steps. I had a look to previous papers from Konrad et al., 2019 and Morris et al., 2017, and I did not really understand how density data were validated before being used in the present paper. My concern is because in Figure 2 we observe that snow density is 550 kg m-3 at 7m below the ground level, whereas it is 600 kg m-3 at the same depth in Morris et al., 2017. Snow density in firn cores from Konrad et al., 2019 are hard compare here because their Figure 3 is developed until 50 m.*

We are afraid that different units for depth, i.e. water equivalent depth (Fig. 2, Morris et al., 2017) and geometric depth (Fig. 2, this study) led to some confusion. Conversion of geometric to water equivalent depth yields an approximate density of $600 \text{ kgm}^{-3}$ at site 21 and 7 m level for this study. Even though, some of the referenced earlier studies use water equivalent scales, we prefer to keep SI units in this study to avoid potential pitfalls from the conversion to a specific reference scale as these are unambiguous. Nonetheless, the difference is well spotted by the referee and highlights the required caution to be taken when comparing different scales.

*Could the author describe whether they calibrated the NP snow density data with snow pit data in the present paper or not? If not, is there a difference in the density/depth relationships between (Konrad et al., 2019), (Morris et al., 2017) and present paper. If snow density from NP measurements is biased, how will this impact the final SMB values?*

Calibration of NP firn and snow density was done independently of the additional core data and based on theoretical considerations. We added a citation of Morris (2008) on line 75, which discusses the calibration equation of NP measurements. In addition, we added the following Line:

"A comparison with gravimetric density measurements at existing core profiles did not indicate a systematic bias between both measurement methods."

**(MC3)** - *Concerning the kriging method, is it worth using northing, easting, and elevation as explanatory variables? Would it be more relevant to use the distance from the coastline as explanatory variable? Or even the distance from Amundsen sea coast?*

Our initial motivation was to follow the methodology by M14 as close as possible for our data comparison. It is an interesting idea, if changes to the parametrisation of the regression model improved the statistical characteristics of residual SMB to the regression surface in our case. However, limiting the explanatory variables to the coastal distance would also limit the degrees of freedom in the regression model. We can only speculate at this point, but orographic effects in response to atmospheric circulation patterns may not be adequately captured by such a parametrisation. We clarified on Line 202 that we do not consider any regression model for our results:

Previous version: "An alternative approach, which is also mentioned in ME14, is a logarithmic transformation [...]"

New version: "We therefore searched for an alternative approach to generate krige estimates from the SMB sample population of this study without the use of a regression model. Such alternative, which is also mentioned in ME14, is a logarithmic transformation [...]"

**(MC4)** - *Comparison of ASIRAS data with RACMO2.3p2 simulations are really interesting but differences are not fully justified/explained in the text. Since differences are model dependant, it would be interesting to see potential differences with another model used in Antarctica (e.g., COSMO-CLM, HIRHAM5, MAR3.10, MetUM , see Mottram et al., 2020). Since Agosta et al. (2019) proposed potential justifications of the differences existing between RACMO2.3p2 and at least the MAR model, I believe that a comparison with the MAR model would make sense here. Indeed, in Agosta et al., 2019, large differences between RACMO and MAR are observed in regions where the RACMO2.3p2 model presents the largest differences with the ASIRAS data. A quick comparison could be relevant to discriminate whether the precipitation formation, advection of hydrometeors, and sublimation of precipitating hydrometeors (Agosta et al., 2019) are important or not in the PIG area. Data from Agosta et al. 2019 are available here: https://zenodo.org/record/2548848#.X0St8TXgphE If the authors are interested in higher resolution simulations, data from a more recent paper from Donat-Magnin et al., 2020, focusing in the Amundsen region are also available at: https://doi.org/10.5281/zenodo.2815907.*

We included simulated SMBs from the high resolution MAR runs by Donat-Magnin, as suggested. We mainly focussed on the model impact on the hybrid total mass input estimates, which is mainly limited to the high elevation interior.

**(MC5)** - *The paper largely describes differences with Medley et al. (2014) paper, but the authors never include any figure presenting the differences. I propose to include a map presenting ME14 route, SMB results and differences with the ASIRAS data.*

We do not have access to the required data, while adding additional maps may affect the conciseness of the manuscript. Our main focus is on the estimates of quantitative SMB properties, which we compare with the study of M14 in the provided tables.

**(MC6)** - *The paper is sometimes quite hard to follow for a non-expert of this area. Different datasets are used here, and the difference between ISTAR/ASIRAS and ME14 is not always clear. The authors use a many acronyms. I suggest that the authors include a table where they clearly describe the difference between ASIRAS/iSTAR data used here, and more particularly the difference with the ME14. For instance, Table 1 presents different radars (ASIRAS, pulseEKKO PRO GPR , CReSIS AccuR) part of the information is given in the caption, but perhaps the authors could also precise in the table if field campaigns were deployed on the ground or by plane,*

We indicate now which radar measurements were performed airborne or groundbased in Table 1.

*over what distance?*

We haven't received flight track coordinates from M14 yet, hence, we cannot make a precise statement in Table 1. Mentioned distances in M14 refer to the combined flight tracks above the Pine Island and Thwaites catchment areas.

*which were the reflection layers used for SMB estimates? which density measurements were considered (NP? Firn cores?)? how was performed the dating of the reflection layers?*

We included the considered SMB averaging periods in Table 1.

*which density measurements were considered (NP? Firn cores?)*

We added the new row "considered density profiles" to Table 1.

*how was performed the dating of the reflection layers?* We added a row to Table 1, which lists the considered annual markers for each study.

**(MC7)** - *Figures could display the route followed by ME14, and where GPR from Konrad et al., 2019 was carried out.*

We still don't have access to flight lines from M14.

*and where GPR from Konrad et al., 2019 was carried out.*

We added the following sentence on Line 65: "The route of these observations closely follows the ASIRAS flight track and are both available at http://gis.istar.ac.uk/ ."

*Minor comments: Abstract*

**(mc1)** - *"Thus there is no evidence of a secular trend in mass input to the PIG basin." => please be more accurate because secular may be misinterpreted here. I suggest to replace secular (here and elsewhere) by decadal.*

We now use "secular trend at decadal scales" in the abstract. "Decadal trend" alone would be more limiting. It is our intention to relate the steady acceleration in ice loss to a potential change in total mass input, i.e does a non-periodic change in total mass input amplify or compensate the mass imbalance of PIG at the observational time scale. Later passages which contain "secular" relate this term to the considered decadal time scales, which should clarify its meaning.

**(mc2)** - *"in particularly"*

Changed to "in particular".

**(mc3)** - *Lines 38-41: firn cores are only used to retrieve the depth of dated snow layer? Are they used to calibrate the neutron probe density data?*

We added on Line 75: " Further details on the calibration procedure, which is based on theoretical considerations, can be found in Morris (2008). A comparison with gravimetric density measurements at existing core profiles did not indicate a systematic bias between both measurement methods."

**(mc4)** - *Figure 1. ASIRAS-iSTAR survey : please also include the location of ground GPR observations from Konrad et al., 2019?*

See our response to MC6. We added the following sentence on Line 65:
"The route of these observations closely follows the ASIRAS flight track and are both available at http://gis.istar.ac.uk/ ."

**(mc5)** - *Line 68, the authors write: " Due to the reported consistency between the GPR and airborne SMB measurements in Konrad et al. (2019), we limit the comparison of our results to the basin wide estimates by ME14". I feel that a figure (perhaps in the supplementary material) showing the different radargrams could help the reader. A quick data comparison, before interpolation, could also be done to see how snow density and radar data uncertainty impact the final SMB value.*

We agree that a comparison of radargrams from different platforms can be useful for an ongoing discussion on the error assumptions made by different studies. However, we are cautious about the expectation that this may be done by means of a quick data comparison as suggested herein. The radar studies of M14 and Konrad et al. (2019) use a different reflection layer for their analysis compared to this study, which already impacts the uncertainty of their SMB estimates. Local noise in the stratigraphy varies along the flight track and impacts related uncertainties in addition. Our error discussion builds upon the developed methodologies in M14 and Konrad et al. (2019), and already considers impacts from the uncertainty in the assumed regional density profile and dating precision for the selected reflection horizons. Figure 9 (b) already gives a benchmark for

other studies to the estimated measurement uncertainty of our results. However, we think that it is of help for potential future studies to add the ASIRAS radargrams, interpolated, and non-interpolated reflection layer picks in units of annual SMB and TWT to the PANGAEA repository.

**(mc6,mc7)** - *Line 85 "Morris et al. (2017) applied an automatic annual layer identification routine to their snow density profiles and used the annual H2O2 peak depths as an additional guidance for the annual layer dating." => is it possible to observe the removal of on year of snow in the PIG area (due to erosion) or multiple summer maxima?*

*Line 88: Here I understand that the authors did not consider the density obtained from the firn cores to compute the final SMB. What is the difference in the final SMB if the authors use the density from firn cores to calibrate the snow density profiles?*

See our response to MC1

**(mc8)** - *Line 89: 43 profiles => I suppose this means that profiles were done twice at the 22 sites?*

Yes, that's correct, but we are now only counting the merged NP profiles as single entities at each site to avoid any confusion about the number of iSTAR sites and density profiles. (See also our response to the second major comment of referee #1).

**(mc9)** - *Line 92: is there any relationship between snow density profile and Accumulation/Temperature as suggested by Herron & Langway 1980 equation?*

we replaced Line 84," Finally we fit an exponential function to the regional mean profile (red dashed line), which we apply to the TWT to depth conversion."

with "Morris et al. (2017) observed a two-stage Herron and Langway (1980) type densification at PIG, with the stages separated by an additional transition zone. We achieve a good fit to our regional mean profile with a simple exponential function (red dashed line, Fig. 2), which we apply to the TWT to depth conversion."

**(mc10)** - *Line 120: how does hoar produce thin ice layers? Is it possible to have short rain our melt events in spring or late summer? Could wind erosion or sublimation create wind crusts in this region?*

According to our cited reference to Arcone et al. (2004) on Line 120, i.e."[...] a hoar layer, which frequently occurs beneath thin ice layers because it is the source of vapor that creates the ice layer", therefore, the answer to this question should already be provided by the cited reference.

*Is it possible to have short rain our melt events in spring or late summer? Could wind erosion or sublimation create wind crusts in this region?*

See our response to MC1.

**(mc11)** - *Line 144: "We assign an estimated date to the mean value of N points" => do you mean "the mean value of depth"?*

Yes, we mean "the mean value of depth". To make our methodology more clear, we rephrased Lines 139 to 147 as follows:

"Based on annual density markers, we can relate the snow and firn depth at each iSTAR site to its associated age and determine the reflection layer age from its depth at each cross section. Here, we use an exponential fit of the local density–depth profile for the TWT to depth conversion. The lateral displacement $\Delta D$ between the point of closest approach of the flight track and iSTAR site adds to the reflection layer dating uncertainty. We therefore consider all $N$ points which lie within a $2\Delta D$ interval along

the flight track and which is centred at the point of closest approach for the layer dating. Based on the local depth–age scale, we relate the estimated depths of $N$ points to their ages and assign the final layer date to their mean value. To account for the mass balance year definition above, we add six months to the mean layer date, which is listed for all iSTAR positions in Tab. 2".

**(mc12)** - *Line 198: "may by due"*

Corrected typo: "may be due"

**(mc13)** - *Line 275: please discuss this sentence according to Agosta et al. (2019) results. Indeed, according to this publication, sublimation of precipitating hydrometeors are missed at low elevation in RACMO2.3p2. This could justify that RACMO 2.3p2 presents a positive bias at low elevation. Conversely, they suggest that MAR snowfall rates generally exceed those simulated by RACMO2.3p2, by more than 30% on the lee side of the West AIS (Marie Byrd Land toward Ross ice shelf), and close to crests at the ice sheet margins. Here, a comparison with MAR could be interesting.*

We included a discussion of the results by Agosta et al. (2019) in the new section "Elevation dependet model drift", which we limited to the overestimation of RACMO results, because of their impact on our hybrid SMB map.

New section: "The observational SMB estimates by M14 indicate an elevation dependent drift of simulated SMB from RACMO. The authors find that RACMO underestimates the SMB at the high-elevation interior, which would also impact our ASIRAS–RACMO based estimates of total mass input. Indeed, this finding is also reflected in our data (see supplement S1) and suggests that the ASIRAS–RACMO based total mass input estimates are biased by the underestimated SMB contribution from RACMO. According to (Agosta et al., 2019), the opposite may apply for the ASIRAS–MAR based estimates. The authors observe a tendency for MAR to overestimate accumulation on Ross-Marie Byrd Land and conclude that differences between MAR and RACMO2 are very likely related to differences in the advection inland. Similar to our elevation dependent comparsion between ASIRAS and RACMO SMB estimates, we find evidence of a drift in the MAR estimates with an opposite sign according to S1. We conclude that the best estimate for total mass input lies between ASIRAS–RACMO and ASIRAS–MAR estimates."

**(mc14)** - *Lines 288: The authors refer to the ASIRAS or the Hydrid SMB estimates, but line 296 they refer to the ASIRAS vs. the ME14 one, whereas they refer to the Hybrid estimate at line 298. The difference between these estimates is not clear. Why do the authors use ASIRAS / hydrid estimates in different parts of the text?*

We revised this section to include the MAR data and clarified our reference to considered hybrid estimates.

Previous version: "The Pine Island $\Sigma_+$ values are in agreement between all data sets within the estimated error margins. This is different for the Wedge outlines, where the RACMO $\Sigma_+$ estimates are between 35-40) % lower compared to the estimates of this study and ME14. Increasing the averaging time of RACMO estimates to the 1985–2009/10 period of the ME14 results yields an increase of $\Sigma_+$ by 2% for the Pine Island and 8% for the Wedge outlines. In comparison with the ASIRAS and ME14 estimates, the simulated total mass input for the Wedge outlines remains outside the error margins of both observations. Considering the further SMB properties according to Tab. 3, the hybrid SMB estimates show the largest variability and range for the Pine Island outlines. This is different for the Wedge outlines, where the hybrid SMB estimates appear to be larger for most of the basin area."

New version: "The Pine Island $\Sigma_+$ values are in agreement between all data sets within the estimated error margins. This is different for the Wedge area, where the RACMO $\Sigma_+$ estimates are between 35–40 % lower compared to the estimates of this study and ME14. Increasing the averaging time of RACMO estimates to the 1985–2009/10 period of the ME14 results yields an increase of $\Sigma_+$ by 2% for the Pine Island and 8% for the Wedge area. However, the RACMO based total mass input to the Wedge area remains below the observational error margins. With regard to the MAR estimates, we find that $\Sigma_+$ values are about 5% higher for Pine Island and 38% higher for the Wedge area. The higher MAR SMB compared to RACMO towards

the southern interior yields a 3% increase for hybrid SMB estimates based on complementary MAR estimates. Considering the additional SMB properties according to Tab. 3, the hybrid based SMB estimates of this study show the largest variability, except for the Wedge area."

**(mc15)** - *Line 325: "but also measured density and strainrate profiles suggest a mean annual SMB of 200 kgm-2yr-1 based on theoretical grounds, which both are in a better agreement with the collocated ASIRAS based results." => Is it possible to explain briefly the way this value of 200kg m-2 is computed?*

We replaced "based on theoretical grounds" with "based on the Herron and Langway stage 1 equation"

**(mc16)** - *Line 352: "secular" => decadal?*

See our response to mc1.

**(mc17)** - *Line 395: "Inspection of the artificial cluster highlighted in Fig. 5 revealed" => replace by fig. 8?*

We replaced "Fig. 5" by "Fig. 8".

**(mc18)** - *Table 2: please include site coordinates and elevation.*

We added latitude, longitude, and elevation to Table 2

**(mc19)** - *Figure 7: Please define NN*

We replaced "NN" with "Rmax=Nearest Neighbours in legend and revised caption: "PP-plots between SMB observations and estimates based on OK and OLK interpolation methods for varying thresholds of their maximum distance Rmax. [...] Average PP-distances (see main text for definition) and SMB values are shown in the legend."

Previous version: "PP-plots between SMB observations and estimates based on OK and OLK interpolation methods for varying maximum estimate distances with regard to the closest measurement locations. [...] Average PP-distances (see text) and SMB values are shown in the figure."

**(mc20)** - *Figure 8: I suggest to include the elevation contours. Is it correct to write Eastings when the x-axis is related to the north-south direction, and Northings for the y-axis when it is on the east/west direction?"*

We included light-toned 200 m elevation contours and replaced axis annotation to Polar Stereographic X and Y to avoid any confusion.

*I don't understand why the kriging procedure induce strange vertical (or horizontal) lines of similar values in Figure 8a (this point is particularly visible near the glacier outlet)? Is it because northing, easting, and elevation are used as explanatory variables? If yes, is it not more relevant to use the distance from the coastline as explanatory variable? Or even the distance from Amundsen sea coast?*

We added the following explanation to Line 259: "Furthermore, some streak artefacts are visible from the interpolation, which are mainly caused by the quadrant criterion of the OLK estimation. Increasing the number of nearest neighbours helps reducing these artefacts but at the cost of PP-agreement in terms of Fig. 7. We therefore kept the OLK settings according to Fig. 8 (a) hereinafter."

---

## Author Response (AR2)

**Author's response to suggested corrections by referees 2**

"The regional scale surface mass balance of Pine Island Glacier, West Antarctica over the period 2005–2014, derived from airborne radar soundings and neutron probe measurements" by S.Kowalewski et al.

*The Cryosphere Discuss.*
*https://doi.org/10.5194/tc-2020-102*

We are pleased to receive the valuable suggestions by referee 2.

Stefan Kowalewski
* * *
In Table 1 : O (10 m) => does it mean approximately 10m?

We are now explicitly writing out „order of" for the "O()" notation, which has been used in the earlier version

In Table 2: I suggest to add the mean SMB value measured at each point because it is interesting for model validation.

Uploaded data to Pangaea contain the alongtrack high resolution and smoothed SMB estimates (Fig. 5a,b). The latter were used for the krige interpolation and may be the favourable choice for comparisons with regional climate models.

Figure 4 a and c : I still don't understand why the combined error in this figure is not the sum of the spatial, temporal and digitization errors.

As stated on line 195, the combined error is calculated by the root-sum-of-squares, which is based on the assumption that individual error components are uncorrelated and normally distributed (Line 190). See also Medley et al., 2013 (supplement), which we refer to on line 199.

Lnie 267 : "By comparison with the flight track shown in Fig. 1, even when considering the practical range as a maximum threshold for the spatial SMB estimation, we do not cover the entire PIG basin." => the boundaries of the spatial SMB estimation are visible in Figure 8 and the authors could refer to this figure here.

We added "see Fig.8" in brackets.

Page 17, Line 320: "With regard to the MAR estimates, we find that sigma+ values are about 5% higher for Pine Island and 38% higher for the Wedge area.." => this sentence is not clear. Do you refer to the comparison between the MAR and M14 or with RACMO? Indeed, the difference between MAR and M14 (for Jul.1985–Jan.2010) is 3% for Pine Island and 18% for Wedge not 5% and 38% respectively.

Here, estimates are compared between RACMO and MAR, which yields the stated percentages. To clarify the comparison, we changed

" With regard to the MAR estimates, we find that sigma+ values […]" to

„In comparison with RACMO and MAR estimates, we find that MAR based sigma+ values [...]"

Table 3 : "RACMO estimates in brackets refer to the July 1985 to January 2010 averaging period in accordance with the results from M14" => there are no values in brackets in the table and I understand from the table that this remark also refers to MAR data? Please verify this caption.

Well spotted! "Brackets" in the caption refers to an earlier version of Table 3. Now, both values are separated by semicolon, which we changed in the caption:

"[…] RACMO and MAR estimates separated by semicolon refer to the July 1985 to January 2010 averaging period in accordance with the results from M14"Appendix A: List of Abbreviations and Notations : This list is really useful. Perhaps the authors could mention this list in the main text ?

We added to Line 53 (end of Introduction): "We include a list of abbreviations and notations in Appendix A"

**Additional minor edits by the authors of this study:**

- Line 195, „Figure 4 (a) displays the propagated individual measurement error components as well as the combined error according to Eq. (5) as a function of geometric depth." -> Added „measurement" between „combined error" to avoid ambiguous usage when speaking in terms of combined errors in the later sections.

  For further clarification, we included two new entries in the abbreviation list:

Combined Error       root-sum-of-squares of measurement and interpolation standard deviation
Measurement Error    root-sum-of-squares of spatial, temporal, and digitization error components

- Line 284, "The horizontal resolution of simulated SMB is 27 km for  RACMO and 10 km for MAR runs.", removed "the" as indicated

- Figure 1, caption: Removed "the" in front of "SCAR Antarctic Database"

- Figure 2. caption: Removed "T1 and" in "Compiled density-depth profiles from traverese  T2 at all 22 iSTAR sites". We noticed that T1 can be confusing in the caption. As stated in the main text, all profiles shown in Fig.2 were collected during T2: Line 79-81

"To evaluate the effect of densification, the ground team repeated the density profiling in the same boreholes during traverse T2. Because the most recent accumulation is missing in these profiles, they drilled an additional borehole of less than 6 m depth and a nearby distance of about 1 m to capture it during traverse T2."

- Figure 8. caption, "Red triangle denotes the position of an interpolation artefact (see  **Sec. 4.5**).", fixed reference to wrong section as incidcated.

- Bibliography, provided URL by Zwally et al. (2012) is now correctly displayed.

- Added to the acknowledgments:
  "We sincerely appreciate the valuable comments and suggestions by the referees and editor."

- Supplement: Added line break before "(c,d)" in the caption

Data availability

Please note that the current reference to the data repository still contains a placeholder:

https://doi.pangaea.de/10.1594/PANGAEA.XXXXXX

According to communication with the PANGAEA support, they are working hard to create the final doi as soon as possible. Hence, we will add the doi number during the final proof phase of the manuscript.